# Evolution and lineage dynamics of a transmissible cancer in Tasmanian devils

Young Mi Kwon[1☯], Kevin Gori[1☯], Naomi Park[2], Nicole Potts[1], Kate Swift[3],
Jinhong Wang[1], Maximilian R. Stammnitz[1], Naomi Cannell[1], Adrian Baez-Ortega[1],
Sebastien Comte[4,5], Samantha Fox[6,7], Colette Harmsen[3], Stewart Huxtable[6],
Menna Jones[4], Alexandre Kreiss[8], Clare Lawrence[6], Billie Lazenby[6], Sarah Peck[6],
Ruth Pye[8], Gregory Woods[8], Mona Zimmermann[1], David C. Wedge[9],
David Pemberton[6], Michael R. Stratton[2], Rodrigo Hamede[4,10], Elizabeth P. Murchison[1]*

1 Transmissible Cancer Group, Department of Veterinary Medicine, University of Cambridge, Cambridge, United Kingdom, 2 Wellcome Sanger Institute, Hinxton, United Kingdom, 3 Mount Pleasant Laboratories, Tasmanian Department of Primary Industries, Parks, Water and the Environment (DPIPWE), Prospect, Tasmania, Australia, 4 School of Natural Sciences, University of Tasmania, Hobart, Tasmania, Australia, 5 Vertebrate Pest Research Unit, NSW Department of Primary Industries, Orange, New South Wales, Australia, 6 Tasmanian Department of Primary Industries, Parks, Water and the Environment (DPIPWE), Save the Tasmanian Devil Program, Hobart, Tasmania, Australia, 7 Toledo Zoo, Toledo, Ohio, United States of America, 8 Menzies Institute, University of Tasmania, Hobart, Tasmania, Australia, 9 Oxford Big Data Institute, University of Oxford, Oxford, United Kingdom, 10 CANECEV, Centre de Recherches Ecologiques et Evolutives sur le Cancer, Montpellier, France

☯ These authors contributed equally to this work.
* epm27@cam.ac.uk

**Data Availability Statement:** Sequence data associated with this study have been deposited in the European Nucleotide Archive with accessions ERP124040 (whole genome sequencing data) and ERP124232 (molecular inversion probe genotyping

## Abstract

Devil facial tumour 1 (DFT1) is a transmissible cancer clone endangering the Tasmanian devil. The expansion of DFT1 across Tasmania has been documented, but little is known of its evolutionary history. We analysed genomes of 648 DFT1 tumours collected throughout the disease range between 2003 and 2018. DFT1 diverged early into five clades, three spreading widely and two failing to persist. One clade has replaced others at several sites, and rates of DFT1 coinfection are high. DFT1 gradually accumulates copy number variants (CNVs), and its telomere lengths are short but constant. Recurrent CNVs reveal genes under positive selection, sites of genome instability, and repeated loss of a small derived chromosome. Cultured DFT1 cell lines have increased CNV frequency and undergo highly reproducible convergent evolution. Overall, DFT1 is a remarkably stable lineage whose genome illustrates how cancer cells adapt to diverse environments and persist in a parasitic niche.

## Introduction

Devil facial tumour 1 (DFT1) is a transmissible cancer affecting Tasmanian devils (*Sarcophilus harrisii*), marsupial carnivores endemic to the Australian island of Tasmania. Spread between devils by the transfer of living cancer cells through biting, DFT1 usually manifests as tumours on the head or inside the mouth [1,2]. Animals with symptoms consistent with DFT1 were first observed in north-east Tasmania in 1996, and the disease has subsequently expanded

data). S1 Data, S2 Data, S3 Data and S4 Data are available in Zenodo with accession 4046235. All other data are contained within the paper and its Supporting information files.

**Funding:** This work was supported by Wellcome grant (102942/Z/13/A) to EPM, https://wellcome. ac.uk/; a Philip Leverhulme Prize from the Leverhulme Trust to EPM, https://www.leverhulme. ac.uk/; National Science Foundation grant (DEB-1316549) to EPM and MJ, https://www.nsf.gov/; Eric Guiler Tasmanian Devil Research Grants to EPM, https://www.utas.edu.au/giving/areas-to-support/research/grants-and-scholarships; Australian Research Council grant (DE170101116) to RH, https://www.arc.gov.au/; and a Herchel Smith Postgraduate Fellowship to YMK, https:// www.herchelsmith.cam.ac.uk/. The funders had no role in study design, data collection and analysis, decision to publish, or preparation of the manuscript.

**Competing interests:** The authors have declared that no competing interests exist.

**Abbreviations:** ANOVA, analysis of variance; AU, approximately unbiased; BAM, binary sequencing alignment map; bp, base pair; CN, copy number; CNV, copy number variant; CN2, copy number 2; DFT1, devil facial tumour 1; DPIPWE, Department of Primary Industries, Parks, Water and the Environment; emmeans, estimated marginal means; GTR, generalised time reversible; IGV, Integrated Genomics Viewer; kb, kilobase; Mb, megabase; MIP, molecular inversion probe; MMLQ, Median minimum base quality for bases around variant; MQ, mapping quality; multipcf, multiple-sample pcf; M5, marker 5; PCA, principal component analysis; pcf, piecewise constant fitting; PON, panel of normals; QD, quality depth; RTK, receptor tyrosine kinase; SC, sequence context; SNV, single nucleotide variant; UMI, unique molecular identifier; VAF, variant allele fraction.

across the island [3,4]. DFT1 is usually fatal and has caused significant declines in devil populations, posing a serious conservation threat to the species [3–5].

Unlike most cancers, which arise from and remain confined within the bodies of their hosts, DFT1 is a clonal lineage derived from the somatic cells of an individual "founder devil." Rather than dying together with this animal, DFT1 continued to survive by spreading as an allogeneic graft through the devil population [1]. Tracing the genomic changes in this long-lived and divergent cell lineage provides an opportunity not only to map the clone's routes through the Tasmanian landscape but also to investigate its underlying evolutionary processes.

Cancer evolves via positive selection acting on somatic genetic changes promoting adaptation to the malignant niche. DFT1 cells have encountered particularly diverse environments during the lineage's expansion, including those of genetically and immunologically variable devil hosts, different tissues within the host including internal metastases, and even artificial conditions such as laboratory cell culture. The exposure of parallel DFT1 sublineages to such environments creates a natural experiment for observing the evolution of cancer cell adaption. In addition, competition between genetically distinct DFT1 sublineages for a limited host population may drive evolution of traits influencing disease transmission, and tracking the interactions and changing distributions of DFT1 sublineages within and across populations could identify clones with competitive advantage. Here, we document the geographical expansion, genomic alterations, and lineage dynamics of the DFT1 outbreak, providing insight into the evolution of a cancer turned infectious disease.

## Results

### DFT1 phylogeny and lineage dynamics

We skim sequenced (approximately 1×) the whole genomes of 648 DFT1 tumours, together with those of their matched hosts, collected throughout Tasmania between 2003 and 2018 (S1 Table). Single nucleotide variants (SNVs) were called individually on each tumour and then assessed across the panel of tumours and hosts (Methods). A total of 456 SNVs showing phylogenetic signal in tumours and absence in hosts were genotyped using molecular inversion probes (MIPs; median 378× coverage, S2 Table) and, together with 94 somatic mitochondrial mutations and 932 copy number variants (CNVs; S2 and S3 Tables), used to construct a DFT1 phylogenetic tree (Fig 1A and S1 Fig).

The DFT1 phylogenetic tree reveals a series of early (pre-2003) nodes, which define five tumour groups, named clades A to E. Two clades, D and E, contain only two and one tumours, respectively, and appear to represent divergent DFT1 sublineages which have not persisted (Fig 1A and S1 Fig). Clades A, B, and C, which together represent 99% of tumours in the sample, are each broadly distributed across Tasmania and show distinctive geographical and temporal patterns (Fig 1B and 1C and S2 Fig).

The distribution of DFT1 clades supports epidemiological evidence suggesting that DFT1 originated in the 1980s or 1990s in north-east Tasmania and first spread rapidly into high-density devil populations in central and eastern areas before a gradual movement towards the more isolated north-west [3,4]. Tumour sublineages that diverged early within all three major clades (A, B, and C) are detectable in the north-east, suggesting that the three clades' most recent common ancestor may have been located there (Fig 1A and 1B and S1 Fig). After divergence from this common ancestor, clade A split into two sublineages, clades A1 and A2, the former spreading south along the east coast and becoming apparently confined to three south-facing peninsulas and the latter colonising central Tasmania and entering naïve host populations at the north-western and southern disease fronts (Fig 1C). Clade B has widespread presence throughout central, northern, and eastern Tasmania, and clade C has followed a westerly

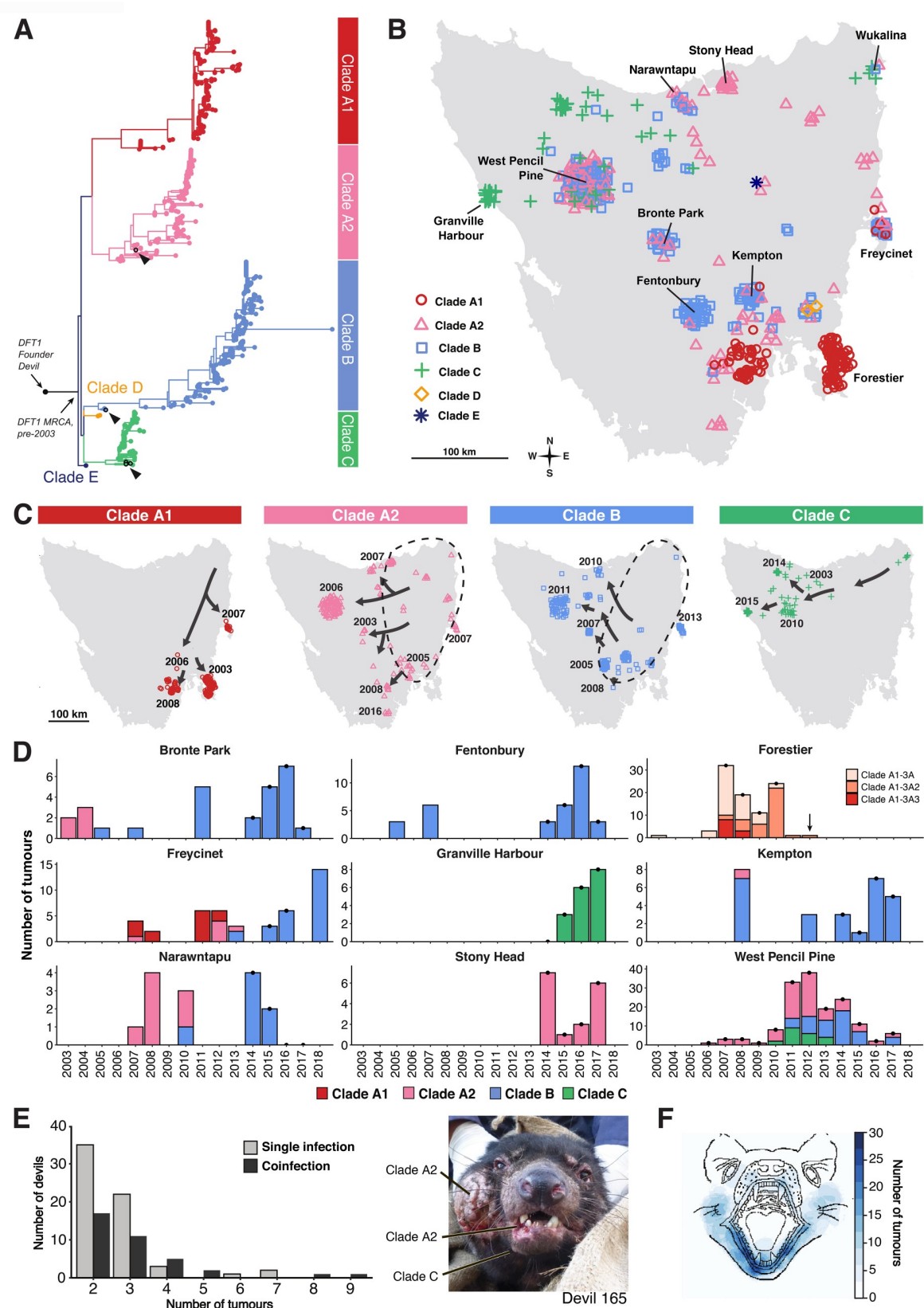

**Fig 1. DFT1 phylogeny and lineage dynamics.** (A) Phylogenetic tree constructed using 550 SNVs and 932 CNVs genotyped across 639 DFT1 tumours (9 tumours were excluded from tree due to missing data). Each tip represents a tumour, coloured by clade. Branch lengths are proportional to number of variants, not evolutionary time, and high-resolution bootstrapped tree is available in S1 Fig. Tumours sampled in the north-east of Tasmania (Wukalina), the putative origin of DFT1, are indicated with arrowheads. SNV and CNV genotypes are available in S2 and S3 Tables, respectively. (B) Map of Tasmania showing distribution of DFT1 clades represented by 593 tumours in (A) and S1 Table (metastases, tumours involving captive devils, and cases of repeated sampling of individual tumours are excluded (S1 Table)). Each tumour is represented by a coloured symbol. Mapping coordinates have been randomly offset in heavily sampled labelled locations in order to aid visualisation. Sample density reflects trapping effort, not tumour prevalence. Tumour clade data and geographical coordinates are available in S1 Table. Map outline was obtained from https://library.unimelb.edu.au/collections/map_collection/map_collection_outline_maps. (C) Spatial spread of DFT1 clades. Each tumour is represented by a coloured symbol. Arrows represent putative spatial movements interpreted from the phylogeny, and year when the DFT1 clade was first observed in each location within the set of sampled tumours is marked. Dotted lines demarcate inferred distributions of clades A2 and B before sampling for this study began, with arrows illustrating inferred tumour migrations out of these core areas. Tumour clade data, sampling dates, and geographical coordinates are available in S1 Table. Map outline was obtained from https://library.unimelb.edu.au/collections/map_collection/map_collection_outline_maps. (D) DFT1 clade distribution in nine locations indicated on map in (B) between 2003 and 2018. Distribution of three clade A1 subgroups is shown in Forestier. Tumour population size reflects sampling effort, not disease incidence, and is not comparable between sites. Years with consistent sampling efforts within sites are marked with black dots. All devils were removed from the Forestier Peninsula in 2012 (arrow). Data are available in S1 Table. (E) DFT1 coinfection. The number of devils with between two and nine independently sampled DFT1 tumours is shown (277 devils had only one tumour sampled, not shown on plot). Those devils for which genotypes of all sampled tumours were indistinguishable or could be distinguished only by variation private to the individual are designated as "single infection." Individual devils hosting tumours with distinct genotypes known to occur in other devils in the population are labelled "coinfection," although in some cases this might represent sampling of the index tumour within which the new variation arose (S4 Table). An image of a coinfected devil (Devil 165) with tumour genotypes labelled is shown on the right. Underlying data are available in S4 Table. (F) Facial distribution of DFT1 tumours. Diagram was constructed by superimposing facial tumour locations of 96 DFT1s. Raw data are available in S1 Data in https://doi.org/10.5281/zenodo.4046235. CNV, copy number variant; DFT1, devil facial tumour 1; km, kilometre; MRCA, most recent common ancestor; SNV, single nucleotide variant.

trajectory, spreading out of the north-east and driving DFT1 towards the remote west coast (Fig 1C). The patterns of connectivity within the devil population revealed by DFT1 lineage tracing reflect host genetic substructure and may be explained by geographical features [6–8].

The interaction between DFT1 clades over time reveals local patterns as well as large-scale trends (Fig 1D). In several populations, clade B has increased in prevalence and replaced clade A (Fig 1D). Sequential lineage replacements have occurred between subgroups of clade A1 (clades A1–3A, A1–3A2, and A1–3A3) isolated on the Forestier Peninsula (Fig 1D), as previously described [9], and may have been due to DFT1 bottlenecks driven by a conservation management trial involving regular removal of diseased animals [10].

DFT1 coinfection, which occurs when individual devils carry genetically distinct DFT1 sublineages from separate exposures, has been observed in devils, but its frequency is unknown [9]. Exactly 100 devils within the cohort hosted two or more sampled DFT1 tumours, and, in 63 of these, the genotypes of these tumours were indistinguishable or could be distinguished only by variation private to the individual (S4 Table). However, tumours in the remaining 37 devils had distinct genotypes, which were also found in tumours affecting other individuals in the population (Fig 1E and S4 Table). Most of this latter set likely represent coinfection, but in some cases, we may have sampled the source of a new genotype. This not only confirms that carrying one DFT1 tumour does not generally protect devils from subsequent infection but also implies that opportunities for within-host competition between sublineages may influence DFT1 evolutionary dynamics [11,12]. To gain insight into the bodily distribution of DFT1 tumours, we superimposed data from 96 DFT1-infected individuals onto a facial map, implicating the lower jaw as the most frequent site of DFT1 tumours (Fig 1F and S3 Fig and S1 Data in https://doi.org/10.5281/zenodo.4046235). Genotyping also identified relationships between primary facial and internal metastatic tumours within individual hosts (S4 Table). Overall, these studies trace the progression of DFT1 over two decades, revealing geographical routes of expansion through Tasmania, patterns of coinfection and metastasis, and the dynamics and persistence of parallel DFT1 strains.

## DFT1 copy number variation

To study genomic heterogeneity within and between DFT1 clades, CNVs were called across the DFT1 tumour cohort using read coverage in 100-kilobase (kb) intervals. We identified 742 CNVs within 567 DFT1 biopsies (CNVs unique to cell lines were excluded), including 315 gains (copy number (CN) >2) and 427 losses (CN <2) (Fig 2A and S3 Table). Most CNVs involved complete gain or loss of a single copy, although some higher amplitude gains, homozygous deletions, and subclonal changes were also observed (Fig 2A and S3 Table). Eight whole genome duplication events [13] were observed across the DFT1 cohort, with some tetraploid lineages persisting through time and others becoming less frequent (S4 Fig). By searching for CNVs common to all tumours, we reconstructed the copy number profile of the most recent common ancestor of the DFT1 cohort, which had a largely unperturbed diploid genome (Fig 2B).

DFT1 gradually acquires CNVs over time, with slight variation between clades and evidence for an early burst prior to 2003 (Fig 2C and S5 Fig). Average diploid DFT1 genome size has remained constant since the emergence of the lineage (S6 Fig). Most DFT1s have a similar number of CNVs; however, several individual tumours carried particularly high CNV burdens, suggesting that they may have undergone sporadic genomic crises or acquired defects in pathways regulating genomic integrity (Fig 2B bottom panel and S1 and S3 Tables). Internal metastases have similar numbers of CNVs to external facial tumours, suggesting that chromosomal instability does not license metastatic spread in DFT1, as has been suggested in some human cancers and model systems (S7 Fig) [14–16]. Some subclonal CNVs were clonal in related tumours occurring in different individuals, suggesting that we sampled the tumour in which the CNV arose (S5 Table). Shared subclones, however, were only detected in two cases of repeated sampling of a single tumour, implying that transmission bottlenecks do not favour the persistence of subclonal cell populations (S5 Table). Overall, these data support the idea that DFT1 is a stable cell lineage that largely maintains its genomic integrity [9,17].

We constructed a genome map of DFT1 CNVs by superimposing gains and losses identified across the tumour population onto devil chromosomes (Fig 2D). This revealed a significant level of CNV clustering (Fig 2D and 2E). Indeed, some genomic loci were independently focally gained or lost 27 times across the DFT1 biopsy cohort (Fig 2D and S3 Table). Some recurrent CNVs were restricted to individual clades, whereas others were identified across clades (S8 Fig).

Recurrent CNVs may occur due to selection, genome fragility and other processes leading to breakpoint clusters, or chromosomal instability. In order to assess selection, we annotated genes occurring in genomic segments that have been independently gained or lost three or more times within focal (<50 megabase (Mb)) CNVs (Fig 2D and S6 Table). This screen identified *PDGFRB*, a receptor tyrosine kinase (RTK) whose amplification likely confers a proliferative advantage in DFT1 [18], as a strong candidate for positive selection (Fig 2D and S6 Table). *PDGFRB* has been independently gained eight times, often at high copy number; interestingly, this has occurred three times in tumours at a single location, the Freycinet Peninsula (S7 Table). Additional recurrent high amplitude focal amplifications were also observed on chromosomes 1, 4, and 5, possibly targeting RTK ligand *NRG2*, anti-apoptosis factor *BIRC5*, and DNA-binding protein *HMGA2*, respectively (Fig 2D and S6 Table). Hemizygous loss of *B2M*, a component of major histocompatibility complex class I, was observed in all DFT1s except for the single ancestral divergent clade E tumour (Fig 2B), presenting the possibility that this event, which likely confers lower immunogenicity [18,19], may have been favoured in early DFT1 evolution.

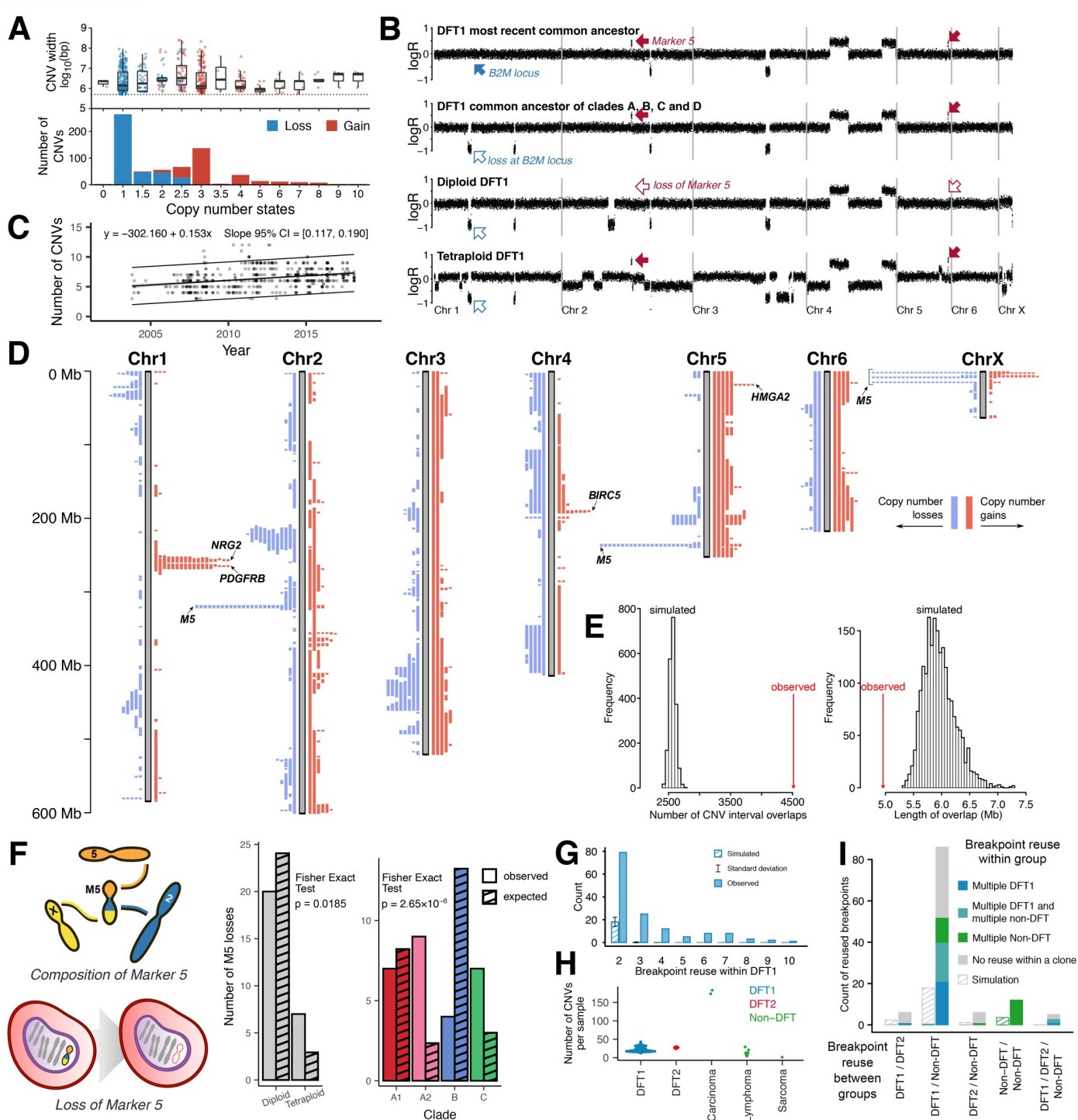

**Fig 2. Copy number variation in DFT1.** (A) Distribution of diploid DFT1 CNV states and widths, measured in bp. Subclonal copy number states are labelled 1.5, 2.5, and 3.5, and represent subclonal states between the two closest integer states. CNVs occurring after a whole genome duplication event are excluded. Gain and loss CNVs with CN2, loss CNVs with CN >2, and gain CNVs with CN <2 represent back mutations. The CNV detection limit was 500 kb (dotted line). Data are available in S3 Table. (B) Representative DFT1 copy number plots. Each dot represents normalised read coverage within a 100-kb genomic window. Tumour identities are, top to bottom, 377T1 (clade E), 56T2 (clade C), 228T1 (clade A2), and 209T3 (clade A1). The diploid tumour, 228T1, has lost M5, whose CNVs are visible on chromosomes 2 and 5 (red arrows). A CNV encompassing major histocompatibility complex class I component *B2M* was acquired in the common ancestor of clades A, B, C, and D (blue arrows). Copy number plots for all tumours are available in S2 Data in https://doi.org/10.5281/zenodo.4046235. (C) Rate of DFT1 CNV acquisition. Each tumour

is represented by a grey dot, and labels mark 1 January of the labelled year. Sets of CNVs cooccurring with the same copy number in the same samples are counted once. CNVs are called relative to the reference genome. Grey shading represents regression standard error, and upper and lower lines represent prediction intervals. We corrected for ploidy in tetraploid tumours, and cell lines are excluded. Data are available in S1 Table. (D) DFT1 CNV chromosome map. Grey bars represent chromosomes with scale on left in Mb. Blue and red bars represent copy number losses and gains, respectively. Each bar is an independent CNV occurrence, and CNVs shared between tumours via a common ancestor are illustrated once. Each CNV step away from CN2 is illustrated separately. Candidate driver genes within frequently amplified regions are annotated, and complete lists of genes within CNV intervals are found in S6 Table. CNVs associated with M5 are indicated. *HMGA2* is not annotated in the reference genome, Devil7.1, but is inferred to be present in the labelled interval based on cross-species genome alignment. CNVs occurring exclusively in DFT1 cell lines are excluded. CNV data are available in S3 Table. (E) DFT1 CNVs are more clustered than expected by chance. Number and length of overlapping CNVs in observed data and in data derived from 2,000 simulations. CNV data are available in S3 Table. (F) Loss of small derived chromosome M5 in diploid and tetraploid DFT1s and across DFT1 clades. M5 was acquired in a DFT1 common ancestor and is composed of fragments of chromosomes 2, 5, and X; this chromosome has been repeatedly lost in the DFT1 lineage (cartoon). The Fisher exact tests compare the observed distribution of M5 losses to those expected by chance, with the latter calculated assuming equal opportunity for M5 loss across all M5–positive tumours, correcting for change in opportunity caused by tetraploidy. Data are available in S8 Table. (G) Frequency of CNV breakpoint reuse within 567 DFT1 biopsies (cell line breakpoints excluded). Ends of chromosomes, M5 CNVs, and instances of reuse within individual samples are excluded from count. Simulated data are derived from 2,000 neutral simulations. Data are available in S9 Table. (H) Number of CNVs in Tasmanian devil cancers of different histotypes. "Lymphoma" includes two unspecified cutaneous round cell tumours (S1 Table). CNVs exclusive to DFT1 cell lines are excluded. Number of CNVs is relative to the reference genome. Each tumour is plotted as a dot, and each non-DFT data point represents an independent clone except for the three dots representing tumours 106T1, 106T2, and 106T3, which are separate tumours belonging to a single non-DFT cancer. Data are available in S1 and S3 Tables. (I) Frequency of CNV breakpoint reuse across Tasmanian devil cancers. Number of reused breakpoints between pairs or trios of devil cancer groups are shown, with colours indicating the number of breakpoints within each category which are also reused within individual devil cancer cohorts. DFT1 cell line CNVs, ends of chromosomes, M5 CNVs, and reuse within individual samples are excluded from count. Simulated data are derived from 2,000 neutral simulations of datasets of the same size. Simulated data are coloured for breakpoint reuse within group using the same colour key as used for real data, as shown on the plot. None of the reused breakpoints were found in the genomes of normal devil tissues, including those of matched hosts (S1 Table). Non-DFT, spontaneously arising non-transmissible devil cancers that are neither DFT1 nor DFT2. Data are available in S9 Table. bp, base pair; CI, confidence interval; CN, copy number; CNV, copy number variant; CN2, copy number 2; DFT1, devil facial tumour 1; LogR, ratio of tumour to normal reads, log base 2; Mb, megabase; M5, marker 5.

Chromosomal instability is characterised by repeated gain or loss of individual chromosomes. The recurrent loss of a cassette of genomic loci involving chromosomes 2, 5, and X, an event which has occurred 27 times within the DFT1 biopsy cohort, represents the loss of "marker 5" (M5), a small derived chromosome present in the DFT1 most recent common ancestor (Fig 2F and S3 and S8 Tables) [13,17]. M5 was previously used to define DFT1 karyotypic strains for lineage tracing [13,17], and our finding that this chromosome is frequently lost suggests that reinterpretation of these data may be required (S1 Table). It seems unlikely that M5 loss is under positive selection, given that it encodes few genes with confirmed involvement in cancer (S6 Table), and its loss returns the majority of its sequence to a diploid state; furthermore, two independent instances of M5 gain were also observed (S8 Table). Rather, M5 may be particularly unstable compared with other chromosomes, perhaps due to an inefficient centromere [20–22]. Alternatively, M5 loss may represent a background rate of chromosome nondisjunction, an event which may be poorly tolerated when involving other chromosomes. Interestingly, M5 has been lost more frequently, relative to opportunity, in tetraploid than in diploid DFT1s, even after correcting for ploidy (Fig 2F and S8 Table). The chromosome has also been retained more frequently in clade B compared with other clades (Fig 2F and S8 Table), suggesting that alterations in M5 structure or cell function have promoted this chromosome's stability within the sublineage.

Some genomic loci are particularly vulnerable to DNA breakage or rearrangement due to processes such as genomic fragility and nonallelic homologous recombination [23,24]. A number of candidate DFT1 sites of genomic instability were identified, including a 12.5-Mb locus on chromosome 5, which was independently gained and lost ten times. A total of 144 genomic breakpoints (at 100-kb resolution) were reused on two or more occasions within the DFT1 lineage (Fig 2D and 2G and S9 Table), although it should be noted that the fragmented state of the reference genome may inflate the number of breakpoint hotspots observed (S3 Table). This contrasted with limited breakpoint reuse in 2,000 neutral simulations (Fig 2G).

Selection and genome instability might also involve common DNA regions in non-DFT1 Tasmanian devil cancers. We investigated this by sequencing genomes from 15 DFT2 tumours (DFT2 is a second Tasmanian devil transmissible cancer that also causes facial tumours [25])

as well as those of ten other spontaneous non-transmissible devil cancers including seven lymphomas and round cell tumours, two anal sac carcinomas, and a single sarcoma (S1 Table). The carcinomas had more than five times more CNVs than other devil cancers, and each harboured high copy number amplicons carrying known cancer genes (*PIK3CA* and *MYCL*, respectively), confirming that CNV mutation burden and cancer gene involvement vary across devil cancer histotypes (Fig 2H and S6 Table). In total, 119 CNV breakpoints (at 100-kb resolution) were reused across two or more independent devil cancers (where DFT1 and DFT2 are each considered independent cancers, and each non-DFT tumour is considered an independent cancer), while few were expected by chance (Fig 2I). In some cases, pairs of breakpoints were reused in the same subsets of samples, leading to CNV recurrence (S9 Fig). Many cross-cancer reused breakpoints occurred more than once within DFT1 and likely reflect common genomic unstable sites (Fig 2I and S9 Table).

## DFT1 cell culture evolution

Laboratory cell lines can be readily established from DFT1 tumours [13,17]. In order to investigate how DFT1 cells adapt to cell culture conditions, we sequenced genomes from 24 cell lines derived from diverse DFT1 tumours across the lineage's phylogeny (Fig 3A and S1 Table). We observed that DFT1 cell lines have significantly more CNVs than DFT1 primary tumours (Fig 3B). In fact, 421 of 1,163 (36%) CNVs identified in DFT1 were unique to cell lines, while cell lines constituted only 6% of samples (S3 Table). Additionally, M5 loss is significantly more frequent in cell lines than in tumour tissue biopsies (Fig 3C). Altogether, this suggests that introduction to cell culture may select for genetically unstable DFT1 subclones, although we cannot exclude the possibility that these observations could result from relaxation of negative selection in DFT1 cell culture.

We generated a chromosomal map displaying the distribution of genomic gains and losses in DFT1 cell lines (Fig 3D). Remarkably, this revealed that the majority of DFT1 cell lines have undergone a series of highly distinctive genomic rearrangements (Fig 3D and S10 Table). These most prominently include losses involving chromosome 4, returning a region gained to three copies early in DFT1 evolution (see Fig 2B) back to two copies and gains involving chromosome 5. These changes were also observed in cell lines derived from tetraploid lineages, and losses and gains of a single copy of regions of chromosomes 4 and 5 were observed in these cases. In four low-passage cell lines, these changes were subclonal, suggesting ongoing evolution in vitro. The phylogenetic distribution of these cell lines on the DFT1 tree confirms that the recurrent alterations do not result from accidental laboratory contamination (Fig 3A). Furthermore, these changes were specific to DFT1, as they were not observed in cultured devil fibroblasts or in DFT2 cell lines (S2 Data in https://doi.org/10.5281/zenodo.4046235). To determine whether the cell line–specific genomic changes were compatible with growth within an animal host, we examined nine tumours derived from sequenced cell lines that had been experimentally introduced into devils [26]. In all of these cases, the cell line genomic configuration was retained in transplanted tumours (Fig 3E).

We annotated the genes present within genomic regions recurrently gained and lost in DFT1 cell lines to identify potential underlying drivers. Surprisingly, *ERBB2*, whose activity is necessary for DFT1 growth [18,27], lies on the arm of chromosome 4 whose third copy is repeatedly lost in cell lines (Fig 3D), indicating that the selective advantage of undergoing this change exceeds the cost of losing a copy of *ERBB2*. Chromosome 5p, which is repeatedly gained in cell lines, carries the ribosomal DNA (rDNA) locus [28], raising the possibility that increased ribosome abundance may offer an advantage to DFT1 cells in vitro (Fig 3D). SNV allele assignment revealed that the chromosomal copies implicated in the events on

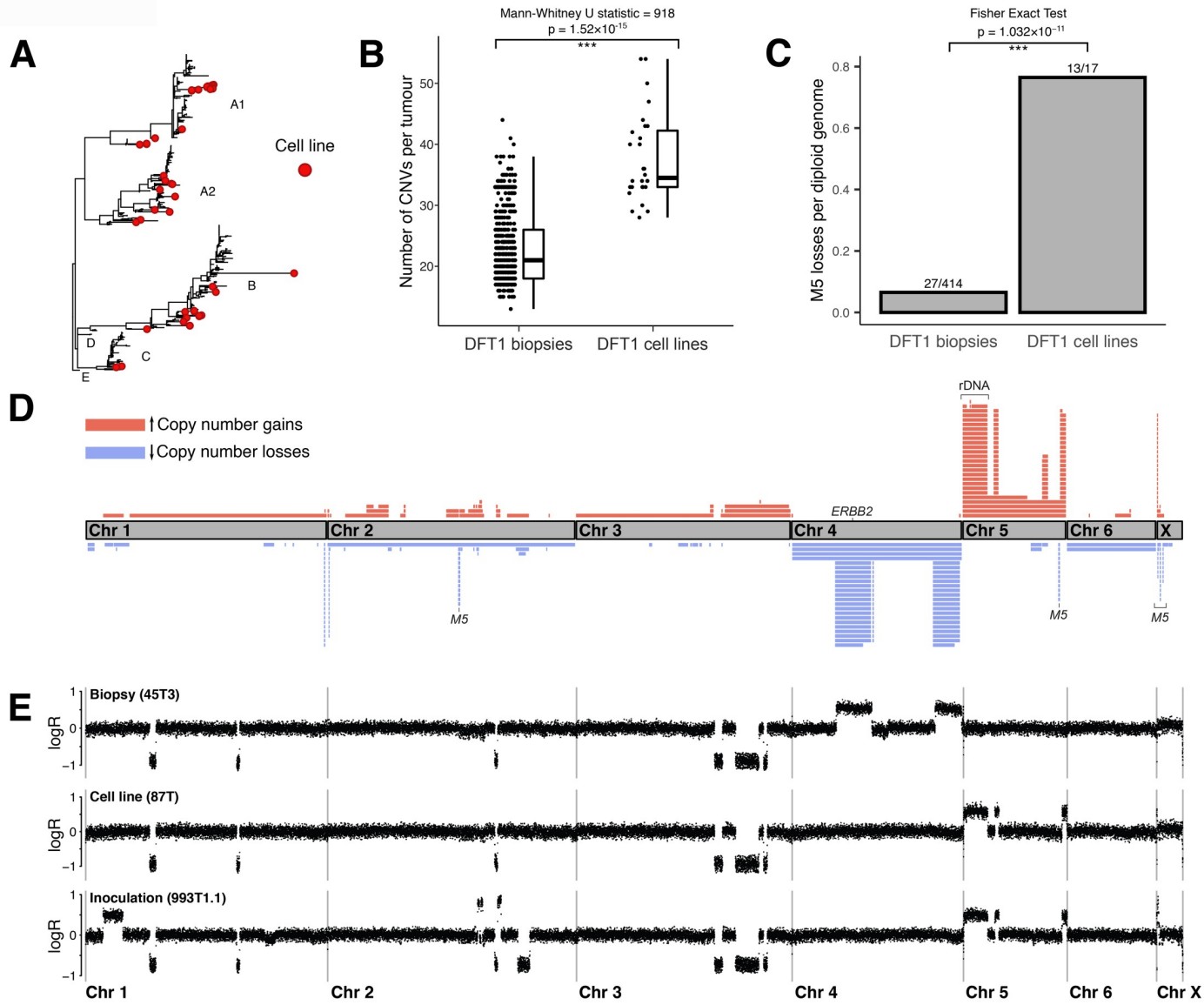

**Fig 3. Copy number variation in DFT1 cell lines.** (A) DFT1 phylogenetic tree with 43 cell lines (including some duplicates collected at different passages), and tissue biopsies from devils inoculated with cell lines, indicated. DFT1 clades are labelled. Higher resolution tree is available in S1 Fig, and underlying SNV and CNV data used to generate tree are available in S2 and S3 Tables, respectively. Sample information, including cell line status, is available in S1 Table. (B) Number of CNVs in DFT1 biopsies and cell lines. Data are available in S1 and S3 Tables. (C) M5 loss events in DFT1 biopsies and cell lines. Cell lines derived from tumour lineages lacking M5 are excluded. Data are available in S8 Table. (D) Chromosome map illustrating CNVs in DFT1 cell lines. Grey bars represent chromosomes. Blue and red bars represent copy number losses and gains, respectively, which were observed in DFT1 cell lines but not in DFT1 biopsies. Each bar is an independent CNV occurrence, and CNVs shared by multiple cell lines via a common ancestor are illustrated once. Each CNV step away from CN2 is illustrated separately, thus a CNV present at CN <1 or >3 is shown multiple times. The location of *ERBB2* is marked, together with a window within which the rDNA occurs (exact coordinates are unknown). Data are available in S3 Table. (E) Representative genomic copy number plots for a DFT1 biopsy, cell line, and experimental inoculation of the same cell line into a devil. Each dot represents normalised read coverage within a 100-kb genomic window. Tumour identifiers are labelled in parentheses, all three samples belong to Clade A1-3A3 (S1 Table). Copy number plots for all tumours are available in S2 Data in https://doi.org/10.5281/zenodo.4046235. CN, copy number; CNV, copy number variant; DFT1, devil facial tumour 1; M5, marker 5; logR, ratio of tumour to normal reads, log base 2; rDNA, ribosomal DNA; SNV, single nucleotide variant.

chromosomes 4 and 5 varied among tumours, indicating that the selective advantage is not allele specific (S10 Fig and S11 Table). Intriguingly, we identified two DFT1 biopsies, one a

lung metastasis, with equivalent losses on chromosome 4 and gains on chromosome 5, suggesting that some DFT1 body sites may mimic cell culture conditions, driving DFT1 adaptation (S11 Fig and S10 Table). Overall, these observations reveal that some cancers respond to altered environments in a highly reproducible manner, indicating that the pathways to achieving fitness optima are, in some cases, surprisingly limited.

## DFT1 telomeres

DFT1 chromosomes have short telomeres compared with those of normal devil cells, but the cancer is known to express telomerase reverse transcriptase *TERT* [29,30] (S12 Fig). To investigate if DFT1 telomere length changes with time, we measured the abundance of telomeric repeats in DFT1 biopsy genomes using tumour purity as a covariate in a linear regression model. This assay suggests that DFT1 telomere length has remained unchanged in tumours sampled between 2003 and 2018 (S12 Fig). These results imply that DFT1 telomeres shortened either before or soon after DFT1 neoplastic transformation; indeed, critically short telomeres may have contributed to the chromosomal rearrangements observed in the cancer [17,31]. Subsequently, telomere lengths have remained constant, probably due to the activity of telomerase.

## Discussion

This survey of DFT1 genomic diversity reveals a stable neoplastic cell lineage that has altered relatively little during its spread through the Tasmanian devil population [9,13,17]. Indeed, more genetic variation, at least at the level of copy number, is frequently found between distinct tumours of a cancer within an individual human patient or even within a single human tumour [15,32–36] than is present in DFT1s sampled from devils hundreds of kilometres and more than a decade apart. These findings suggest that early in its evolution, at least at the common ancestor of clades A, B, C, and D, DFT1 acquired a genomic configuration compatible with growth within a range of environments, including those of genetically and immunologically variable devil hosts and internal metastatic sites.

Nevertheless, despite exceptional adaptation to its niche, there is some evidence of ongoing positive selection in DFT1. Repeated focal and high amplitude copy number gains involving the *PDGFRB* locus suggest selection for increased cell proliferation [18,37]. Surprisingly, given evidence for strong selective pressures exerted by the host immune system [38], we did not identify obvious recurrent genomic changes contributing to immune evasion. Such adaptations may not be copy number driven or may involve poorly assembled or annotated portions of the reference genome. However, the potential for DFT1 to adapt to novel conditions is clear from the convergent structural changes observed in DFT1 cell lines. Given the declining density of devils, particularly in eastern and central Tasmania [4], as well as evidence for high frequency of DFT1 coinfection, there is scope for DFT1 inter-lineage competition. Our observation that clade B has largely remained behind the disease front and repeatedly replaced other DFT1 sublineages raises the possibility that this clade may have a selective advantage, perhaps, for example, favouring transmission in post-disease low-density devil populations, although we cannot exclude the possibility that these dynamics are stochastic. Additionally, repeated acquisition of *PDGFRB* copy number gains in tumours of different clades on the Freycinet Peninsula, coinciding with steady devil population decline [39], suggests the possibility of an arms race towards more virulent DFT1 forms driven by within-host inter-lineage competition. These findings contrast with the absence of detectable ongoing positive selection in the 6,000-year-old canine transmissible venereal tumour [40], perhaps reflecting differences in age and within-host inter-lineage competitive dynamics between the two cancers.

Although positive selection continues to mould the DFT1 genome, most genetic alterations in DFT1 are unlikely to offer an advantage. Indeed, the gradual accumulation of mutations may, in the long term, reduce the lineage's robustness, especially given evidence that negative selection may be inefficient in transmissible cancers [40]. Any potential costs of continued clonal evolution are, however, unlikely to affect the immediate consequences of DFT1 on its precarious host population.

DFT1 presents a continued and serious threat to the Tasmanian devil, and predicting the future impact of the disease remains challenging [5,41–43]. The spatial and temporal dynamics of DFT1 between 2003 and 2018, described here, reveal not only the trajectories of parallel and competing DFT1 sublineages but also trace the patterns of movement of the diseased devils themselves. In combination with findings from ongoing population monitoring studies [4,38,39,44–46], these data may be useful for epidemiological modelling and prediction of management intervention benefit. The recent emergence of DFT2 [25,45] further exacerbates the species' peril and underlines the importance of conservation efforts.

Overall, we have described the phylogeny, lineage dynamics, geographical spread, and genome evolution of DFT1, providing important information for future Tasmanian devil conservation planning and illustrating how a comparatively simple and stable cancer cell lineage can colonise diverse niches and devastate a species.

## Methods

### Ethics statement

Animal procedures and sample collection followed a standard operating procedure that was approved by the General Manager of the Natural and Cultural Heritage Division in the Tasmanian Government Department of Primary Industries, Parks, Water and the Environment (DPIPWE), in agreement with the DPIPWE Animal Ethics Committee and covered by a Scientific Permit issued by the Tasmanian Government, or were approved under University of Tasmania Animal Ethics Permits A009215, A0011436, A0013685, A0012513, A0014976, A0010296, A0013326, and A0016789 with State Scientific Permit TFA 19144. Sample collection procedures were approved by the University of Cambridge Department of Veterinary Medicine Ethics and Welfare Committee (CR191).

### Sample collection and DNA extraction

Biopsies were collected from wild or captive Tasmanian devils or from animals whose carcasses were found as roadkill. Tumour and host (ear, liver, spleen, blood, gonad, or submandibular lymph node) tissue biopsies were collected into either RNAlater (Thermo Fisher Scientific, Waltham, United States of America) or ethanol. Fine needle aspirates were collected into a 1:1 solution of Buffer AL (Qiagen, Hilden, Germany): PBS were sampled from tumours that were too small to biopsy. Cell lines were established and cultured as previously described [13,17,18]. Experiments involving inoculation of devils with DFT1 cell lines have been described [26]. Genomic DNA was extracted using the Qiagen DNeasy Blood and Tissue Kit (Qiagen). Samples included in the study, as well as metadata for identifying samples used in various analyses, are listed in S1 Table. Cell lines used in the study are also known by the following external identifiers: 85T (Half-Pea); 86T, 86T2 (1426); 87T, 87T2, 90T, 90T2, 114T, 363T1, 993T1.1, 993T1.2, 994T1.1, 995T1.1, 995T1.2, 996T1, 996T2 (C5065); 88T, 88T2 (4906); 204T (Ed); 202T2 (RV, TD467); 203T3 (SN, TD500). Groups of samples belonging to the same parental cell line were sampled at different passages or after inoculation into devil hosts (details in S1 Table).

## Sequencing

Illumina whole genome sequencing libraries were generated with an average insert size of 450 base pairs (bp) using standard methods according to manufacturer's instructions, and were tagged with Sanger index tags for multiplexing. We pooled and sequenced approximately 17 samples per lane on an Illumina HiSeq instrument with version 4 chemistry (Illumina, San Diego, USA) in high output mode with paired-ends (2 × 125 bp) to achieve a median coverage of 1× per sample. Reads were aligned to the Devil7.1+MT Tasmanian devil reference genome (http://www.ensembl.org/Sarcophilus_harrisii/Info/Index) [9] using BWA-MEM (http://bio-bwa.sourceforge.net/bwa.shtml) version 0.5.9-r16+rugo with options "-l 32 -t 6." PCR duplicates were flagged using Picard (http://broadinstitute.github.io/picard/).

## Note on Tasmanian devil chromosome nomenclature

Two Tasmanian devil chromosome nomenclature systems are in use, which differ in the naming of the two largest chromosomes [1,17]. In both systems, the two largest chromosomes are named 1 and 2, but the labels for these chromosomes are swapped with respect to each other. The Devil7.1+MT reference genome [9], used here, follows the system established by Pearse and Swift [1].

## Variant calling and genotyping

**Mitochondrial variant calling and filtering.**   Mitochondrial DNA was sequenced to a median depth of 275×. Mitochondrial variant calling and genotyping was performed with Platypus (version 0.8.1) [47]. First, we called SNVs on each sample individually using default settings with "—minRead = 3" and "—minPosterior = 0." We merged variants and ran Platypus a second time across all samples. We then removed any variants with flags "badRead," mapping quality ("MQ"), or quality depth ("QD").

We applied the following post-processing filters to the remaining variants:

- *Regions filter*: Variants mapping within 500 bp from the start or end of the mitochondrial contig were excluded.

- *Low Variant Allele Fraction (VAF) filter*: Variants observed with VAF <0.2 were discarded, as they were considered to be likely due to low-level cross-contamination.

- *Germline filter*: Variants appearing in both tumour and matched host were discarded as likely germline contamination. In tumours without matched hosts, variants that had been discarded in ≥1 matched tumour were discarded.

We visually validated remaining variants using Integrated Genomics Viewer (IGV) version 2.3 [48].

Mitochondrial DNA (mtDNA) purity was estimated as the mean VAF of tumour-specific mtDNA variants (S1 Table).

**Mitochondrial haplotype nomenclature.**   We created an mtDNA haplotype nomenclature. Each name begins with a prefix identifying the clone: The DFT1 clone is denoted as "DFT1_," and DFT2 is denoted as "DFT2_." A hierarchical notation is then used, which follows the mitochondrial tree topology, alternating between numbers and letters (S1 and S2 Tables).

**Nuclear variant calling and filtering.**   Scaffolds from each chromosome in the reference genome Devil7.1+MT (http://www.ensembl.org/Sarcophilus_harrisii/Info/Index) were concatenated to produce chromosome-level coordinates for each chromosome (S12 Table).

SNV calling and genotyping were performed with Platypus version 0.8.1 [47] with options "—minRead = 1," "—minBaseQual = 30," "—badReadsThreshold = 30," "—filteredReadsFrac = .8," "—badReadsWindow = 15," "—minPosterior = 0," and "—minFlank = 0." For the initial calling, we used Platypus with the parameters described above to identify SNVs in each DFT1 separately. Variants flagged by Platypus as badReads, MQ, strandBias, Median minimum base quality for bases around variant (MMLQ) <30, and sequence context (SC) were removed. For the variant set called in each sample, we applied the following filters:

- *Regions filter*: We excluded variants mapping within 500 bp of the start or end of a contig and within 1,000 bp from the start and end of a scaffold. Variants called in scaffolds not assigned to chromosomes and in mtDNA were also removed.

- *Simple repeats filter*: Variants called within 5 bp from a simple repeat region as annotated by Tandem Repeat Finder [49] were removed.

Each tumour's SNV set was then genotyped in its respective matched host (or a single host, 100H, if matched host was unavailable, S1 Table), and variants identified in host were excluded. We then merged remaining variants into a single file and genotyped this across all tumours and normals using Platypus with default settings but retaining the "—minReads = 1" option. Variants flagged with MQ or badReads were excluded, and we retained those variants with support (≥1 reads) in DFT1s but without support (0 reads) in normal devils.

## Preliminary phylogenetic tree inference and MIP probe selection

We generated a custom sequence template that concatenated Devil7.1+MT to only include sites where variants were observed. The IUPAC ambiguity character (N) was substituted at sites where there was no read coverage across the variant position in a particular sample. We combined the sequences of genotyped tumours and the reference to generate a sequence alignment for phylogenetic tree inference. We constructed a preliminary maximum likelihood tree rooted with the reference sequence, using the generalised time reversible (GTR) substitution model with a gamma model of site heterogeneity with Lewis ascertainment bias correction implemented in RAxML [50]. Using DFT1 mtDNA haplotypes and the preliminary nuclear SNV tree, we placed tumours into preliminary phylogenetic groups. The SNV set was further filtered based on these groups in order to select phylogenetically informative SNVs. A total of 600 SNVs were selected for genotyping using MIPs.

## MIP genotyping

We designed custom MIP capture probes with a 120-bp target capture size using MIPgen [51]. The common linker sequence CTTCAGCTTCCCGATATCCGACGGTAGTGT was replaced with CTTCAGCTTCCCGATCCGCTCTTCCGATCT to enable amplification with Sanger iPCR indexed primers (96-Plex Molecular Barcoding for the Illumina Genome Analyzer https://link.springer.com/protocol/10.1007%2F978-1-61779-089-8_20). We selected 600 probes which passed the recommended scoring thresholds (>1.4 using the support vector regression model and 0.7 using the logistic regression model). The oligos were column synthesised and purified using standard desalting (Integrated DNA Technologies, Coralville, USA). We first pooled the probes at equimolar concentration. We phosphorylated the pooled MIP probes using T4 Polynucleotide Kinase (New England Biolabs, Ipswich, USA) with conditions detailed in Table 1 with incubation at 37˚C for 45 minutes followed by heat inactivation at 65˚C for 20 minutes.

Genomic DNA was diluted to 3 ng/μL, and phosphorylated pooled MIP probes were diluted in EB buffer (Qiagen) to enable a reaction ratio of MIP probes to genome copies of

**Table 1. Conditions for MIP phosphorylation reaction.**

| | Volume (µL) |
|---|---|
| MIP pool (500 µM total concentration) | 20 |
| T4 PNK | 10 |
| T4 PNK buffer (10×) | 7 |
| ATP (10 mM) | 7 |
| Nuclease-free water | 63 |
| Total | 100 |

ATP, adenosine triphosphate; MIP, molecular inversion probe; T4 PNK, T4 polynucleotide kinase.

800:1. The hybridisation and circularisation reactions were performed as shown in Table 2 using Ampligase (LGC Biosearch Technologies, Hoddesdon, United Kingdom) and Hemo Klentaq (New England Biolabs).

DNA was denatured at 98°C for 3 minutes. Temperature was decreased incrementally (1°C per second) to 60°C, held for 1 hour, and the reaction incubated overnight at 56°C. To select for the circularised product, we performed an exonuclease treatment (Exonuclease I and Exonuclease III (New England Biolabs)) with the conditions described in Table 3.

The capture reaction was incubated in the exonuclease mix at 37°C for 40 minutes followed by 95°C for 2 minutes to heat inactivate the exonuclease. We then amplified (Q5 High-Fidelity 2X Master Mix, New England Biolabs) our selected product with the MIP forward primer (5′-AATGATACGGCGACCACCGAGATCTACACATACGAGATCCGTAATCGGGAAGCTG AA*G-3′) together with iPCR indexed primers with the conditions specified in Tables 4 and 5.

Library quality was analysed directly on an Agilent 2100 Bioanalyzer System (Agilent Technologies, Santa Clara, USA) using the Bioanalyzer DNA 1000 Broad Range chip. Libraries were pooled (5 µL of each library) and size-selected (>150 bp) using Ampure beads (Beckman Coulter, Brea, USA) at 0.8×. The purified pooled library was eluted in 22-µl buffer EB (Qiagen) and assessed using a Bioanalyzer High Sensitivity chip (Agilent Technologies).

We first performed a pilot with 96 samples, sequencing the libraries on a MiSeq (Reagent Kit v3 600-cycle, Illumina) with 150 bp paired-end reads. We demultiplexed and trimmed adaptors using bcl2fastq2 v2.20 (https://support.illumina.com/downloads/bcl2fastq-conversion-software-v2-20.html), allowing for no mismatches ("—barcode-mismatches 0"). We then aligned the reads using BWA-MEM (http://bio-bwa.sourceforge.net/bwa.shtml) to Devil7.1+MT. Using a customised script based on Allelecount (https://github.com/cancerit/alleleCount) with min-base-qual = 30, we identified the total number of reads covering the

**Table 2. Conditions for MIP hybridisation and circularisation reactions.**

| | Volume (µL) per reaction |
|---|---|
| Ampligase 10× buffer | 1.3 |
| Phosporylated MIP pool (1:1,000 dilution) | 0.484 |
| dNTPs (0.1 mM) | 0.08 |
| Hemo Klentaq | 0.08 |
| Ampligase (5 U/µl) | 0.1 |
| Nuclease-free water | 0.01 |
| DNA (30 ng) | 10 |
| Total | 12 |

dNTP, deoxynucleoside triphosphate; MIP, molecular inversion probe.

**Table 3. MIP exonuclease reaction conditions.**

|  | Volume (μL) per reaction |
|---|---|
| Ampligase 10× buffer | 0.1 |
| Exo I | 0.125 |
| Exo III | 0.125 |
| Nuclease-free water | 0.65 |
| Total | 1 |

Exo, exonuclease.

region of the SNV, the number of reads supporting each allele, and the number of reads carrying a 5-bp unique molecular identifier (UMI). We calculated the on-target ratios of each probe by calculating the number of unique reads that mapped to the correct genomic region as a proportion of the total number of reads that carried either or both the extension and ligation sequences of each probe. We removed overrepresented probes that were greater than 1.5 log (10)-fold from the overall median UMI read count across samples and probes, and we removed probes with no representation. We also removed probes that carried greater than 50% median off-target ratios across the samples. The MIP probe concentration for rebalancing was calculated by taking the square root of the median coverage over the median coverage for the individual MIPs. Remaining probes were rebalanced and pooled using an Echo 550 Liquid Handler (Labcyte, San Jose, USA). Using the rebalanced MIP probes, we remade and sequenced the new libraries for the same set of 96 samples. From the analysis, we observed that two probes were over 10-fold higher than the overall median UMI read count across probes. We halved the volume of these two probes for the input pool using the same rebalanced concentration. We repooled the remaining MIP probes and genotyped across the samples (S2 Table).

Several tumours were sequenced in duplicate, and in these cases, data were merged from the two sequencing runs. We assessed the number of reads supporting the reference and variant alleles at each target site as detailed above (S2 Table). We then genotyped variant presence or absence as follows:

*Calculate the expected number of reads for presence*: The expected number of reads supporting a variant for a diploid site carrying a heterozygous somatic variant was calculated as follows:

$$Expected\ reads = Total\ reads \times purity \times 0.5$$

Purity was defined as the median VAF*2 for variants that pass a minimum threshold of variant read support. The accuracy of this approach was validated by visualising VAF histograms for each sample.

**Table 4. Conditions for MIP amplification and index incorporation.**

|  | Volume (μL) per reaction |
|---|---|
| Q5 High-Fidelity 2X Master Mix | 13 |
| MIP F primer (100 μM) | 0.13 |
| Indexed MIP R primer (10 μM) | 1.3 |
| MIP capture reaction | 12 |
| Total | 26.4 |

MIP, molecular inversion probe; F primer, forward primer; R primer, reverse primer.

**Table 5. Thermal cycling conditions for MIP amplification and index incorporation.**

| Temperature | Time | Cycles |
| --- | --- | --- |
| 98˚C | 30 s | 1× |
| 98˚C | 15 s | 23× |
| 58˚C | 30 s | |
| 72˚C | 30 s | |
| 72˚C | 2 min | 1× |

*Genotyping*: Genotyping was then performed as follows:

*Expected number of reads ≥10*:

- If the observed number of reads supporting a variant was greater than the contamination threshold (a value calculated individually for each tumour based on the level of cross-contamination (S2 Table)), the variant was considered present.

- If the number of reads supporting a variant was less than the contamination threshold, but >2 reads, the variant was assigned an ambiguity code "N."

- If the number of supporting reads was ≤2, the variant was considered absent.

    *Expected number of reads <10 but ≥5*:

- If the number of supporting reads was greater than the contamination threshold, the variant was considered present.

- If the number of reads supporting the variant was less than the contamination threshold, the variant was assigned an ambiguity code "N."

    *Expected number of reads <5*:

- The variant was assigned an ambiguity code "N."

We excluded variants observed to be present in normal tissue from one or more devils from downstream analyses. We further pruned the filtered set of variants by excluding any variant for which more than 50% samples were assigned the ambiguity code "N." We additionally removed any tumours where more than 50% variants were assigned the ambiguity code "N."

## Copy number calling

For autosomes, we counted read depth in adjacent 100-kb genomic bins across Devil7.1+MT, excluding the last bin of each scaffold as well as short (<100 kb) and unplaced scaffolds. Additionally, the short scaffolds placed towards the end of each autosome were excluded, as specified in [18]. For chr X, which is enriched for short scaffolds, we first concatenated scaffolds prior to bin counting and did not exclude the last bin of each scaffold. Scaffolds beyond position chrX:54800000 (S12 Table) were excluded. Bin counts were normalised by log2 transforming the ratio of read depth within each bin to median autosomal read depth in the panel of normal (PON) genomes (LogR). Normal devil genomes were pooled into a PON and used to denoise tumours using tangent principal component analysis (PCA) normalisation (https://gatk.broadinstitute.org/hc/en-us/articles/360037593691-DenoiseReadCounts). LogR plots (tumours), raw read depth plots (normal devils), and raw read count per bin (tumours and normal devils) are available in S2–S4 Data in https://doi.org/10.5281/zenodo.4046235, respectively.

## Purity calculation

Tumour purity was calculated using copy number features as follows. We estimated purity ($\rho$) using average read depth calculated in 10-kb intervals with a 5-kb slide across the genome ($r_{genome}$) and genomic regions that have lost either 1 ($r_{LOH}$) or both ($r_{Homdel}$) copies in all DFT1 or DFT2 tumours. These regions were identified by assessing copy number states through nonoverlapping 10-kb windows across targeted scaffolds (Table 6). The following formula was used, where n and k represent the number of homdel and LOH regions used to estimate purity, respectively.

$$\rho = \frac{\sum_{i=0}^{n} \frac{\bar{r}_{Homdel,i}}{\bar{r}_{genome}} + \sum_{i=0}^{k} \frac{\bar{r}_{LOH,i}}{\bar{r}_{genome}} - \frac{1}{2}}{n + k}$$

See S12 Table for equivalence of scaffold names. Purity was also estimated from MIP genotypes, as described above in section "MIP genotyping." The mean of both methods was used in downstream analyses. Purity estimates obtained from both methods are available in S1 Table.

**Copy number segmentation.** LogR profiles were segmented using both piecewise constant fitting (pcf) and multiple-sample pcf (multipcf), implemented in R [52]. multipcf was run on groups of tumours with shared mtDNA haplotypes (S1 Table). If pcf and multipcf segments within a single tumour shared a copy number state (see below) and overlapped across >95% of their length, they were merged, and multipcf breakpoints retained. The union of pcf and multipcf segments was further filtered by merging adjacent segments whose read depth dispersions were not significantly different ($p$ = 0.05) using Ansari–Bradley and Wilcoxon tests (https://rdrr.io/bioc/aCGH/). Non-copy number 2 (CN2) segments called in different tumours were merged if they overlapped across >95% of their length and had breakpoints within 500 kb of each other. In these cases, breakpoints were consolidated to the median position of observed breakpoints. Finally, segments <500 kb were discarded. Each tumour's segments were visually inspected and validated (S2 Data in https://doi.org/10.5281/zenodo.4046235). Breakpoints were screened for absence in the set of normal devil genomes (S3 Data in https://doi.org/10.5281/zenodo.4046235).

**Copy number assignment.** The copy number associated with each segment was estimated using two models: a Bayesian hierarchical model implemented in JAGS (http://www.stats.ox.ac.uk/~nicholls/MScMCMC15/jags_user_manual.pdf) and a Poisson logit-linked general linear model implemented in R.

*Bayesian hierarchical copy number assignment.* The read depth of a segment ($r_s$) can be estimated by the sample's ploidy ($\psi$) and purity ($\rho$), defined as the proportion of tumour cells in a sample, as well as the segment's copy number state (Eq 1) [53]. We developed a Bayesian hierarchical model with a Markov chain Monte Carlo-based Gibbs sampler implemented in JAGS

**Table 6. Coordinates and deletion class for scaffolds used in purity calculation.**

| Type | Deletion class | Scaffold coordinates |
|---|---|---|
| DFT1, DFT2 | LOH | Chr3_supercontig_000000288:1090000- Chr3_supercontig_000000297:40000 |
| DFT1 | LOH | Chr2_supercontig_000000346:1- Chr2_supercontig_000000353:269000 |
| DFT1 | Homdel | Chr2_supercontig_000000278:490000–510000 |
| DFT2 | LOH | Chr1_supercontig_000000015:1- Chr1_supercontig_000000027: 440000 |
| DFT2 | LOH | Chr5_supercontig_000000095:1- Chr5_supercontig_000000109:650000 |
| DFT2 | Homdel | Chr3_supercontig_000000242:130000–190000 |

DFT, devil facial tumour; Homdel, homozygous deletion; LOH, loss of heterozygosity.

(http://www.stats.ox.ac.uk/~nicholls/MScMCMC15/jags_user_manual.pdf) to fit the read depths of segments in each tumour and to obtain estimates of a tumour's purity and ploidy and calculate copy number states. For DFT1 tumours, we estimated purity through DFT1-specific regions of deletions in the genome and MIPs data (see above). We used the average of these tumour purity measurements to initialise the tumour purity parameter of the model. Rearranging the formula in Eq 1, we then used the optimised tumour purity and ploidy estimates and the read depths of the segments to assign copy number to the segments (Eq 2).

$$r_s = log_2 \left( \frac{2(1 - \rho) + \rho(CN)}{\psi} \right) \qquad (1)$$

$$CN = \frac{(\psi 2^{r_s} - 2(1 - \rho))}{\rho} \qquad (2)$$

*Poisson model for copy number assignment.* As a cross-validation measure, we obtained the idealised logR for copy number states 1 to 9 using the tumour purity and ploidy estimates using Eq 1 and generated a Poisson mixture model implemented in R to compute the probability that given its read depth, a segment will be of a particular copy number state. The base assumption in this model is that copy number state (n) scales linearly with read coverage (k) (Eq 3). To calculate expected read coverage states for this model, we first calculated the average read coverage of the segments that were assigned as CN2 by the previous copy number assignment model ($\tilde{k}$). We then inferred the expected read coverage of the other copy number states, multiplying the predicted read coverage by the tumour purity ($\rho$) to account for host contamination. We assume that the read coverage distribution within copy number states is under a log-gamma distribution. Using the Poisson mixture model, we then assigned each segment to the copy number state that had the highest probability (Eq 4).

$$r_n = \frac{\tilde{k}_{CN2} n \rho}{2} \qquad (3)$$

$$P(n) = e^{(k \, \log(r_n) - r_n - lgamma(k+1))} \qquad (4)$$

Discrepancies between copy number assignments using the two methods due to low sequencing quality and subclonality were manually validated based on raw read coverage. To further remove false positive breakpoints, adjacent segments that were assigned the same copy number state were merged. Copy number states in all tumours were visually inspected and validated (S2 Data in https://doi.org/10.5281/zenodo.4046235) and mtDNA and MIP phylogenetic trees were used to identify and split recurrent CNVs and to infer back mutations.

Whole genome duplication was inferred using karyotype information (S1 Table), together with observations of CNV copy number states.

## Phylogenetic tree

The phylogenetic tree of DFT1 tumours was inferred jointly from three sources of sequence data: SNVs of the nuclear genome derived from MIP sequencing, mtDNA SNVs and binary presence-absence information corresponding to the CNVs observed in each sample. CNVs associated with M5 and high copy number amplicons were excluded from tree inference. The data were analysed using IQ-TREE 1.6.12 [54], under an edge-linked proportional partition model (-spp option) [55], in which each of the three data sources formed a subset of the partition, to which was assigned its own substitution model. The substitution model for each subset was chosen using IQ-TREE's model selection procedure (-m MF option) [56], with the

following results: MIP SNVs were analysed using K2P+ASC+R2, mtDNA SNVs were analysed using TIM2e+ASC, and CNVs were analysed using GTR2+FO+ASC+R3. Branch support was assessed using IQ-TREE's ultra-fast bootstrap [57] with 1,000 replicates (-bb 1,000 option). The devil reference sequence (Devil7.1+MT) was used as an out-group to root the tree.

Two tree search strategies were employed in the phylogenetic analysis: one in which all candidate trees must conform to a topological constraint (using the -g option of IQ-TREE) [58] and an unconstrained tree search. The topology constraint used was to enforce the samples' preassigned membership to clades A1, A2, B, C, D, and E, based on mtDNA haplotypes and presence of clade-specific SNVs from low coverage whole genome sequencing data (this was relevant for samples missing MIPs data (S1 Table)), as well as to group samples 332T and 471T1, which share an mtDNA haplotype, but 471T1 is missing MIPs data. We ran 20 independent runs for each strategy. We used the approximately unbiased (AU) test [59] to identify any trees that were statistically significantly worse than the maximum likelihood tree among the set. The highest likelihood constrained tree not rejected by the AU test is presented in Figs 1 and 3 and S1 Fig.

## Clade group assignment

We assigned tumours to 37 "clade groups" (S1 Table and S1 Fig) based on position in a phylogenetic tree constructed from MIP and mtDNA genotypes. These were designated with a numeric suffix after specifying the clade (A1, A2, B, and C), e.g., Clade_A1-1, Clade_A1-2, etc. One clade B cell line, 204T, lacked MIPs data and could not be confidently placed within a group; it was therefore assigned its own group, Clade_B-204T. Clade groups and phylogenetic position within the MIPs+mtDNA tree were used to identify CNV recurrence and back mutation.

## CNV rate and genome size variation rate

The accumulation of CNVs with time was plotted using the R package ggplot2. Data were analysed using a linear model; concretely, the lm() function and prediction intervals were inferred by the predict() function implemented in the R programming language. Models were inferred for the relationship between the number of CNVs and the sampling date and for the number of CNVs and the interaction between the sampling date and the DFT1 clade. The significance of the interaction with clade was assessed using analysis of variance (ANOVA), through the Anova() function in the R package "car," and contrasts between pairs of clades were assessed using the emtrends() function in the R package estimated marginal means ("emmeans"). Sets of CNVs which cooccur with the same copy number in the same samples are counted once (see Total_CNVs_linked.clonal_copynumbers_only in S1 Table). We corrected for ploidy in tetraploid tumours (CNVs occurring after whole genome duplication were counted as half a CNV, as there is twice as much DNA in which CNVs can occur after this event). Exact sampling dates were unavailable for 23 samples, although the year of sampling was recorded; for these samples, a random date from the year of sampling was used when calculating the regression.

Diploid DFT1 genome size rate of change with time was analysed as follows. Genome size change was calculated for each tumour by subtracting the "total genome affected loss" from the "total genome affected gain" where "total genome affected gain" and "total genome affected loss" are the sum of the widths of the copy number gain and copy number loss CNVs present in each sample, respectively (S1 Table). Genome size per sample is calculated as 2*haploid genome size + genome size change, where callable haploid genome size was estimated as 2,625,758,988 bp (this excluded regions of the genome which were excluded from copy

number analysis, see above). The rate of change of diploid DFT1 genome size due to copy number variation over time was estimated by robust regression, implemented in the R package MASS, in the function rlm. The "Genome size" response was regressed on the sampling date for each sample. Non-DFT1s, duplicate samples from the same tumour, cell lines, and tetraploid tumours were excluded from this analysis, as were CNVs with subclonal copy number states. Exact sampling dates were unavailable for 23 samples, although the year of sampling was recorded; for these samples, a random date from the year of sampling was used when calculating the regression.

## Allelic copy number

We estimated allelic copy number by intersecting CNV and MIP coordinates. CNV–MIP combinations were only retained if the MIP alternative allele was either present in the tumour(s) carrying the CNV (gains) or inferred to be present in an ancestor tumour (losses). CNV–MIP combinations were further filtered for those that overlapped with 2 or more independent CNVs to assess whether specific alleles are involved in recurrent CNVs.

Each MIP position has an associated estimated total copy number and an observed VAF, and each sample has an estimated purity; these values are used to estimate the genotype. For a given total copy number, we construct an array of potential genotypes, in half-integer intervals, to test. For example, for a total copy number of 2, we would test the genotypes 0.0|2.0, 0.5|1.5, 1.0|1.0, 1.5|0.5, and 2.0|0.0, where the genotype is written in the format "R|A," with R meaning "reference allele" and A meaning "alternative allele." We calculate the expected VAF for each genotype as

$$VAF = \frac{H_{Alt}(1-p) + T_{Alt}p}{H_{Total}(1-p) + T_{Total}p},$$

where $H$ and $T$ represent the copy number in the host and tumour, respectively, and the subscripts $Ref$, $Alt$, and $Total$ designate that the copy number refers to the reference or alternative allele or is the total copy number, and $p$ is the sample purity. For the MIP positions under consideration, $H_{Alt} = 0$ and $H_{Ref} = 2$. The genotype which minimised the absolute difference between expected and observed VAF was selected as the best fit allele-specific copy number genotype at the MIP position for the sample (S11 Table).

To further validate the finding that any copy of chromosome 4 can be lost in the back mutations of CNVs 3R1 and 9 commonly observed in DFT1 cell lines, we performed capillary sequencing of four variant loci occurring within the footprints of CNVs 3R1 and 9 (S11 Table). All four SNVs were inferred to be present in the DFT1 most recent common ancestor. The following primers were used to amplify these loci as approximately 400 bp amplicons in whole genome amplified DFT1 biopsy or cell line genomic DNA:

Chr4_supercontig_000000112:154498 forward: GAGAAGAAAGGGGAGAAGAAATACTAC

Chr4_supercontig_000000112:154498 reverse: AATATAAGGCTACCCACAAACTACCAC

Chr4_supercontig_000000265:1474441 forward: GAAAACTCTTCTGGTTACCATGTACTC

Chr4_supercontig_000000265:1474441 reverse: GGAAAATAGAGAAAGAAGAGTCCTACC

Chr4_supercontig_000000109:783087 forward: CTACTATCATACAGAATACCTGCTGGA

Chr4_supercontig_000000109:783087 reverse: GTCTTCCTCAAATATAACCTCAGACAC

Chr4_supercontig_000000268:868897 forward: GACCTGATCTTAGTAAGGAACTAGGAA

Chr4_supercontig_000000268:868897 reverse: ATTCACAGAACTTAGATTGAGTACCCC

PCRs were performed in 40-μl reactions with the following reagents: 8-μl 5× Phusion HF Reaction Buffer (New England Biolabs), 4-μl DNA template, 0.8-μM dNTP mix (New England Biolabs), 0.8-μM forward primer, 0.8-μM reverse primer, and 2.5-U Phusion HF DNA Polymerase (New England Biolabs). Thermal cycling was performed as follows: 3:30 minutes at 94˚C, then 30 cycles of 30 seconds at 94˚C, 30 seconds at 60˚C, 30 seconds at 72˚C, followed by 5 minutes at 72˚C using an Eppendorf thermal cycler (Eppendorf, Hamburg, Germany). Genotypes were called manually by viewing electropherograms using Geneious R9 software (Biomatters, Auckland, New Zealand).

## Simulated CNV distributions

We assessed whether the observed overlap of independent CNVs was statistically greater or less than expected by chance. We sampled random regions from the genome that matched the sizes of the observed CNVs identified in DFT1s and assessed their overlaps. We performed 2,000 permutations to simulate the null hypothesis that the occurrence of observed CNVs in our set was not associated with genomic position. For the simulation, the genome was circularised in numerical order such that the end of a chromosome is connected to the start of the next, higher number chromosome; the end of chromosome X is linked to the start of chromosome 1. For each observed CNV, the start position was randomly resampled from all available genome positions, and the end position of the CNV was updated to a position downstream of the new start position such that the original CNV length was maintained.

## Visualisation of clade spread

We used a customised R script using the R package "ggplot2" to overlay a plot of the coordinates of each sample to a map of Tasmania (https://library.unimelb.edu.au/collections/map_collection/map_collection_outline_maps) to assess the spread of each identified clade.

## Telomere repeat abundance analysis

Telomere repeat abundance was estimated from binary sequencing alignment map (BAM) files using a modified version of telseq version 0.0.1 [60], using merged read groups and a read length of 125, appropriate for the sequencing technology used (-m -r 125 option). We modified parameters in the telseq source code to adapt it for the devil genome, which include read length, total GC content (bp), and number of chromosomal ends. The number of chromosomal ends was specified as 1 in order to obtain a measure of overall telomere repeat abundance rather than an estimate of telomere length. The modified source code is available at https://github.com/TransmissibleCancerGroup/telseq.

The relationship between the estimated telomere length and the year of sampling was estimated using a generalised linear model, using a log-linked gamma distribution to model errors and to reflect that the length estimates can only take positive values. The implementation used was the glm() function in the R programming language. "Telomere length" estimated by telseq is the cumulative length of telomere repeats and will include interstitial telomere repeats.

### DFT1 face maps

DFT1 tumour images from confirmed DFT1s (S1 Table) were used to construct DFT1 face and body maps. Facial and body tumour contours from photographs or field notes were hand drawn onto devil face and body templates using Krita software (Krita, Deventer, the Netherlands) on a Wacom Intuos Pro tablet (Kazo, Japan). We generated a custom python script using "matplotlib" and "opencv" packages to stack the tumour contour diagrams and to quantify the density of tumour occurrence across the devil face and body (Fig 1F and S1 Data in https://doi.org/10.5281/zenodo.4046235).

### Computer code

Custom software and code associated with this study are available in the repository (https://github.com/TransmissibleCancerGroup/TCG_2020_devil_paper).

## Supporting information

**S1 Fig. DFT1 phylogenetic tree.** Phylogenetic tree of 639 DFT1 tumours, including cell lines, constructed with 456 MIP genotypes, 94 mtDNA mutations, and 932 CNVs; this is a higher resolution version of the tree shown in Fig 1A. Each tumour is labelled with its ID, location and year of sampling, and clade group (S1 Table). Nine tumours (8T2, 9T, 120T, 166T4, 390T1, 460T1, 478T2, 1070T1, and 1072T1) were excluded due to substantial missing data; these tumours' clade groups can be found in S1 Table. The 26 tumours labelled in italic font have missing data or very low purity, and their exact positions on the tree may be inaccurate; in these cases, likely true phylogenetic position is presented with the labelled clade group and in S1 Table. Bootstrap values from 1,000 replicates are reported for each node. Some nodes (base of clades A, B, C, and D and that linking samples 332T and 471T1) were constrained, and as a result, bootstrap values are shown as 100. Branch lengths are proportional to number of mutational events, not evolutionary time. Sample information is available in S1 Table, and SNV and CNV genotypes are available in S2 and S3 Tables, respectively.
(PDF)

**S2 Fig. DFT1 sample locations by year.** Locations of 593 DFT1 tumours collected between 2003 and 2018, coloured by clade. These are the same data as shown in Fig 2B, but split by year. Map outline was obtained from https://library.unimelb.edu.au/collections/map_collection/map_collection_outline_maps. Underlying data are available in S1 Table.
(PDF)

**S3 Fig. DFT1 tumour body locations.** Locations of confirmed DFT1 tumours illustrated on Tasmanian devil (A) face, right-hand side, (B) face, left hand side, and (C) body diagrams. Scale represents number of tumours. The number of individuals in the sample with tumours within the relevant body area were 36, 32, and 32, top to bottom. Underlying data are available in S1 Data in https://doi.org/10.5281/zenodo.4046235.
(PDF)

**S4 Fig. DFT1 tetraploid lineages.** Clade and temporal distributions of the eight observed DFT1 biopsy tetraploid lineages. The years of first and most recent observations of each lineage are indicated. Each lineage is labelled with a numeric ID. Lineage 8 was observed only in a cell line and is not included. Data are available in S1 Table.
(PDF)

**S5 Fig. CNV rates by clade.** Rate of CNV acquisition in DFT1 clades. X-axis represents collection date, with labels representing 1 January of the labelled year. Sets of CNVs which cooccur

with the same copy number in the same samples are counted once, and we corrected for ploidy in tetraploid tumours. DFT1 cell lines were excluded. The difference in rate between clade C and the other clades is statistically significant ($p < 0.005$, pairwise contrasts of emmeans, with Tukey multiple testing adjustment). Coloured shading represents regression standard error, and upper and lower lines represent prediction intervals. Data are available in S1 Table (Total_CNVs_linked.clonal_copynumbers_only) and S3 Table.
(PDF)

**S6 Fig. DFT1 diploid genome size variation with time.** Diploid genome size was estimated for each tumour by adding and subtracting CNV gain and loss footprints from the diploid Tasmanian devil callable genome size (haploid size, 2,625,758,988 bp). X-axis represents collection date, with labels representing 1 January of the labelled year. Subclonal CNVs, tetraploid tumours, duplicate tumours, and cell lines were excluded. Each tumour is represented by a dot, and shading represents regression standard error. Robust regression was performed, $p$-value for association = 0.3849 (robust F-test). Data are available in S1 and S3 Tables.
(PDF)

**S7 Fig. CNV burden in DFT1 primary tumours and metastases.** Number of CNVs does not differ between DFT1 biopsies collected from primary facial tumours ($n = 543$) and internal metastases ($n = 24$). Each dot represents a tumour. Data are available in S1 Table.
(PDF)

**S8 Fig. Clade-specific CNV maps.** Chromosome map illustrating subsets of CNVs in DFT1 biopsies (A) shared by all DFT1s; (B) exclusive to and shared by all DFT1s in clades A, B, C, and D; (C) exclusive to clade E; (D) exclusive to clade A1; (E) exclusive to clade A2; (F) exclusive to clade B; (G) exclusive to clade C; (H) exclusive to clade D. Grey bars represent chromosomes, with scale on left in bp. Blue and red bars represent copy number losses and gains, respectively. Each bar is an independent CNV occurrence, and CNVs shared by multiple tumours via a common ancestor are illustrated once. Each CNV step away from CN2 is illustrated separately, thus a CNV present at CN $<1$ or $>3$ is shown multiple times. Data are available in S3 Table.
(PDF)

**S9 Fig. Maps for intersecting CNVs between devil cancers.** Chromosome map illustrating recurrent CNVs with identical breakpoints (at 100-kb resolution) occurring within one or more tumours belonging to each cancer group in each of the subsets: (A) DFT1 and DFT2; (B) DFT1 and non-DFTD; and (C) DFT2 and non-DFTD. Non-DFTD, non-devil facial tumour disease, i.e., neither DFT1 nor DFT2. Grey bars represent chromosomes, and gains and losses are represented by coloured bars above and below the chromosomes, respectively. The colour key identifies the tumours whose CNVs are represented by each colour. Three non-DFTD tumours (106T1, 106T2, and 106T3) belong to the same non-DFTD cancer sampled from 3 body sites and CNVs were collapsed (106Tx). No CNVs occurring independently in more than one non-DFTD tumour, or in one or more DFT1, DFT2 and non-DFTD tumour were found. This figure refers to recurrent CNVs between devil cancers, and differs from Fig 2I, which describes reused single breakpoints. Underlying data are available in S3 Table.
(PDF)

**S10 Fig. Allelic copy number.** CNV chromosome map displaying CNVs from DFT1 biopsies and cell lines whose footprint encompassed genotyped MIP alleles (S2 Table and S3 Table). MIP genotypes and CNVs are only shown if the CNV depth in a single direction (gain, loss) was $\geq 2$ and if the MIP ALT allele was either present in the tumour(s) carrying the CNV

(gains) or inferred to be present in an ancestor tumour (losses). Gains are coloured red and losses blue, and chromosome coordinates in bp are shown in scale bar on left. The allele (REF, ALT) implicated in each CNV is shown and CNV IDs (S3 Table) are labelled. CNVs 3R1 and 9, parental CNVs of the recurrent chromosome 4 back mutations observed in cell lines, are shown three times each in order to illustrate informative MIPs genotypes that were captured in different samples. The figure shows only directly observed genotypes, although in some cases additional genotypes can be imputed. Lines connect CNVs that cooccur on the same chromosome in the same direction (gain, loss) in the same tumour(s) and which may be linked. None of the stacked genotyped CNVs reached the minimum threshold (9) needed to obtain sufficient power to detect allelic skew with a Fisher exact test. Data are available in S11 Table.
(PDF)

**S11 Fig. DFT1 biopsies with similar CNV profiles to cell lines.** Genomic copy number plots for two DFT1 tumours, a lung metastasis (134T6 and a tumour biopsy of unknown body location (likely facial) (428T1)), with CNV profiles similar to those seen in DFT1 cell lines. The changes observed in 428T1 occurred after whole genome duplication. Copy number plots from the primary facial tumour from the same animal as the lung metastasis (134T1), and a representative cell line (214T), are also displayed. Each dot represents normalised read coverage within a 100-kb genomic window. R, ratio of tumour to normal reads, log base 2. Tumour identifiers are labelled in parentheses. Copy number plots for all tumours are available in S2 Data in https://doi.org/10.5281/zenodo.4046235.
(PDF)

**S12 Fig. DFT1 telomeres.** (A) Telomeric repeat length in DFT1, DFT2, and normal devils ("hosts"). Each sample is represented by a dot. Five outliers in the "host" group are not shown on the plot but were included in the data used to generate boxplot. Data are available in S1 Table. (B) Linear regression of telomere repeat length and year of collection for DFT1 biopsies with tumour purity included as a covariate. Residual errors are plotted in red, and blue shading represents the standard error of the gradient estimate. In both (A) and (B), telomere length is the cumulative length of telomere repeat (TTAGGG) detected in the genome, including both telomeric and interstitial repeats, which cannot be distinguished using this method.
(PDF)

**S1 Table. Sample information.** (A) Tumour sample information. Identifiers and metadata are listed for DFT1, DFT2, and non-DFTD tumours (non-DFTD refers to tumours which are neither DFT1 nor DFT2). Each row refers to a single tumour, with multiple samples from the same individual devil linked by a common individual ID and a digit following the "T," e.g., 31T1 and 31T2 are two tumours sampled from the same host. Further information on devils with multiple tumours can be found in S4 Table. In some cases, "T1" and "T2" refer to the same tumour sampled twice, either at the same or different time points; these are labelled as "duplicate DNA extraction from the same tumour" (same time point) or "Time-course" (different time points). Cell line samples are also labelled "T," and samples from the same cell line collected at the same or at different passages are labelled "T1" and "T2," with those collected at the same time point labelled "Duplicate cell pellet." In addition, the following group of cell lines and experimental transmissions all derive from the same parental cell line (87T, 87T2, 90T, 90T2, 114T, 363T1, 993T1.1, 993T1.2, 994T1.1, 995T1.1, 995T1.2, 996T1, 996T2). Cell lines used in the study are also known by the following external identifiers: 85T (Half-Pea); 86T, 86T2 (1426); 87T, 87T2, 90T, 90T2, 114T, 363T1, 993T1.1, 993T1.2, 994T1.1, 995T1.1, 995T1.2, 996T1, 996T2 (C5065); 88T, 88T2 (4906); 204T (Ed); 202T2 (RV, TD467); 203T3

(SN, TD500). "Matched" and "unmatched" refer to availability of host (see (B)). Clade and clade group (S1 Fig) are listed for each tumour, as well as mitochondrial haplotype group and genotype (S1 Fig and S2 Table). Tumour purity estimates from copy number analysis, MIP genotyping, and mitochondrial analysis are shown. Tumour data, including sampling location, latitude, longitude, date of sampling, host age, and sex (M, male; F, female), are provided. "Singleton" and "multiple" denote individuals from which either a single tumour or multiple tumours were sampled as part of the set, and information on tumour body location is provided, when known. Tumours sampled from an internal body location are marked "metastasis." Information on the diagnosis of non-DFTD tumours is provided. Cell lines and tumour biopsies derived from experimental transmission trials involving either cell lines or tissue biopsies are marked [26], with inoculation date provided, if known. Tumour strain refers to cytogenetic analysis [13]. Subsets of samples included in the phylogenetic tree (Fig 1A and S1 Fig) and in geographical analyses (Fig 1B–1D and S2 Fig) are indicated, including those that are italicised in S1 Fig. Subsets of samples for which different data types are available (mtDNA haplotype, CNVs, and MIPs genotypes) are listed. Samples which are duplicate biopsies sampled from the same tumour at the same time, duplicate cell pellets from a single cell culture, or which are samples from the same tumour or cell line collected on different dates (time course) are indicated. M5 genotype is provided, with blank indicating that M5 is retained. Tetraploid tumours are marked, and the tetraploid lineage to which each tetraploid tumour belongs is listed in parentheses (S4 Fig). CNV burdens and the width of genome affected (bp) are presented, with separate data for all CNVs and clonal CNVs only. CNV strings are concatenated lists of CNV IDs (S3 Table) and "linked CNVs" groups together those CNVs that cooccur in the same direction (gain, loss) in the same samples and which may have occurred as a single event (S3 Table); CNVs present in a sample that do not form part of a linkage group are included in this count as singleton groups. Fig 2C and S5 Fig show data for "linked CNVs," with CNVs occurring after whole genome duplication counted as half a CNV in order to correct for genome size. Subclonal CNVs could not be called in tetraploid tumours and the "clonal only" fields have been left blank in these tumours. Telomere length data, presented in S12 Fig, are available in column labelled "Telseq_Length_Estimate." "SangerStudyIDs" contains additional sample identifiers. (B) Normal devil sample information. Sample ID, microchip (if known), trapping location, gender, and year of sampling are indicated. If the individual is a matched host from a tumour in the study it is marked "matched," otherwise, it is labelled "unmatched." Availability of WGS and MIP data are marked. Telomere length data, presented in S12 Fig, are available in column labelled "Telseq_Length_Estimate." "SangerStudyIDs" contains additional sample identifiers.
(XLSX)

**S2 Table. MIP and mitochondrial genotypes.** (A) MIP genotype expressed as VAF. The number of reads supporting the variant and the number of reads covering the variant site are shown for each MIP–tumour combination. MIP probe sequences are shown, "N" within the MIP probe sequence denotes the 5-bp UMI. (B) Tumour genotypes across the MIP panel. N, genotypes that could not be determined. MIP probe sequences are shown, "N" within the MIP probe sequence denotes the 5-bp UMI. (C) Contaminating read threshold, determined empirically for each tumour, used in genotype calling (see Methods). (D) Mitochondrial (mtDNA) genotypes. Numerals "1" and "0" represent variant presence and absence, respectively. (E) mtDNA haplotype nomenclature. DFT1 mtDNAs are arranged into 36 haplotype groups, based on the presence of haplogroup-defining variants. Haplotypes falling within defined groups acquiring additional variants are named hierarchically with letters and numbers, as shown. Haplogroups DFT1_18 and DFT1_30, as well as haplogroups DFT1_22 and DFT1_29,

are each defined by the same variant (10620T>A and 16143G>A, respectively). The single DFT2 haplotype is also listed. DFT1_0 and DFT2_0 are identical to the reference genome. Small insertions and deletions (indels) are not included.
(XLSX)

**S3 Table. Copy number variants.** The table has one row per CNV, with coordinates, width and type (gain, loss) shown. Each CNV is assigned a basic ID and an extended ID. The basic numeric ID is unique to the pair of coordinates and type (gain, loss), regardless of phylogenetic relationship and may be present in multiple rows (with the exception of partial back mutations, which share the same CNV ID as the parent CNV, but whose coordinates are within the parent CNV footprint). The extended ID is unique per row and incorporates additional information with the following suffixes: R (recurrence, a repeated phylogenetically unrelated CNV involving the same coordinates and type; each recurrence is numbered), BM (back mutation, a CNV which returns a phylogenetically earlier CNV to CN2 or another lower amplitude state), PBM (partial back mutation, similar to BM, but the CNV coordinates lie within the parent CNV footprint), G (gain, higher amplitude gain beyond CN3; each independent gain is labelled, i.e., G1, G2, etc., and the copy number state is also labelled, i.e., G1CN4), and L (similar to G, but for losses below CN1). The samples and clade groups of samples carrying each CNV are listed in the "Sample_group" column, together with the copy number state assigned. Subclonal copy number states are expressed as a 0.5 step between the two closest integer states. "Copy_number" lists each state found in samples carrying the CNV, and the number of samples with copy number at each state are shown. In cases where back mutations have occurred, final copy number state is presented also in the parent CNV, including in cases where the back mutation is partial. CNVs that occur uniquely in cell lines or in cell lines experimentally inoculated into animals are tagged, and "Cancer_type" specifies if the CNV occurs in DFT1, DFT2, or non-DFTD tumours. The "subclonal" tag marks CNVs that have non-integer copy number; however, these may also include integer copy number states in tetraploid tumours (tagged). CNVs that occur exclusively within tetraploid tumours are marked, together with the whole genome duplication (WGD) event number in parentheses (S4 Fig); CNVs that are inferred to have occurred in a common ancestor of a tetraploid tumour group prior to WGD are marked; others are inferred to have occurred after WGD. Linkage groups tag groups of CNVs that cooccur in the same direction (gain, loss), and CNVs that involve the same coordinates but the opposite direction (gain, loss) are marked. Nested CNVs are CNVs that have arisen within the coordinates of a "parent" CNV and are in the same direction (gain, loss) as the parent. The total number of tumours carrying each CNV is listed in the "Number_of_tumours" column. CNVs that were part of the M5 complex or which were part of high-level amplicons were not used to generate the phylogenetic tree (Fig 1 and S1 Fig) and are tagged in the "Do_not_use_-to_build_tree" column. We have tagged intervals that include physically map markers (labelled "Deakin_2012_markers" and "Taylor_2017_markers") [17,31]. The number of tumours carrying each CNV as a fraction of the total number of tumours within each clade group is shown, as well as individual genotypes for all tumours in the set ("1," presence; "0," absence). Copy number plots labelled with CNV IDs can be found in S2 Data in https://doi.org/10.5281/zenodo.4046235.
(XLSX)

**S4 Table. Tumour genotypes in devils with multiple tumours.** Only DFT1 tumours were considered in this analysis, and DFT1–DFT2 coinfection, which occurred in two devils (Devil-812 and Devil-818) in the cohort, is not included. (A) Summary of tumour genotype information for devils hosting more than one sampled tumour. Each animal has an individual ID, and individual tumours sampled from the devil are labelled "T1," "T2," etc. "Type" lists various

categories of multiple tumour sampling regimens: "multiple tumours," when several tumours were sampled from an individual devil at a single sampling time and "time-course," when the same tumour was sampled more than once on different dates. Individuals from which internal metastases were sampled are tagged "metastases collected," and the tissue location of the relevant tumour(s) is tagged. Each animal is marked as "same" or "different," depending on if its tumour genotypes are distinguishable using the range of markers that we have assessed (mtDNA and CNVs). Those animals that have different genotypes between multiple tumours are further classified as "polymorphic public," i.e., all genotypes identified in the animal are shared with tumours in other devils, and "polymorphic private," i.e., any new variation in the devil's tumours is found only in that devil. The genetic distance between different tumours within a single host with polymorphic public genotypes are listed and can include coinfection with tumours of different clades, different clade groups, or the same clade group. Each genotype within a devil is given a number, and these are listed in the "pattern" column. Additional metadata, as described in S1 Table, are also provided. (B) Genotype information from cell lines sampled multiple times at different passage or after inoculation into animals. Data are organised as in (A). (C) Summary of data in parts (A) and (B), with underlying data for Fig 1E.
(XLSX)

**S5 Table. Shared subclones.** (A) Details of subclonal CNVs identified in two or more DFT1 biopsies. Subclonal CNVs that were identified in a single tumour are not included; CNVs that were subclonal in some tumours but clonal in others are included. Upper table, CNVs excluding M5; lower table, M5 subclonality. Linkage group refers to a set of CNVs that cooccur with the same copy number in the same sets of samples and which likely occurred as a single event, with numbers referring to CNV IDs (S3 Table). (B) Details of subclonal CNVs identified in two or more DFT1 cell lines or biopsies from tumours derived from cell lines experimentally inoculated into animals. Table structure as in (A).
(XLSX)

**S6 Table. Gene annotation.** Genes wholly or partially present within genomic intervals covered by CNVs to a specified depth within different subsets of tumours. Interval coordinates are presented, and "Depth" represents the cumulative coverage across the interval in CNV gains or losses (specified in "Type"). The CNV IDs of associated CNVs are shown (S3 Table). The "COSMIC" and "COSMIC summary" columns annotate genes within intervals that are included in the COSMIC Cancer Gene Census [61] (accessed 18 November 2019), as well as COSMIC annotation for these genes. (A) DFT1 biopsies (DFT1 cell lines excluded), genomic intervals covered by at least 3 gain CNVs or at least 3 loss CNVs. *HMGA2* is not annotated in the reference genome Devil7.1, but its presence is inferred in interval 5:17100000–18299999 is based on alignment with other species' genomes. (B) DFT1 cell lines, genomic intervals covered by at least 3 gain CNVs or at least 3 loss CNVs. (C) DFT2, genomic intervals covered by at least 2 gain CNVs or at least 2 loss CNVs. (D) Non-DFTD tumours, genomic intervals covered by at least 2 gain CNVs or at least 2 loss CNVs.
(XLSX)

**S7 Table. Geographical locations of tumours with *PDGFRB* copy number gains.** CNV IDs (see S3 Table) and associated copy number states for gain CNVs encompassing the *PDGFRB* locus. Tumour identifiers, clades and clade groups, and copy number state within each tumour are shown. The number of tumours carrying each CNV and the locations and years in which these tumours were sampled are listed. CNV127 was sampled in 5 tumours, 4 located in Freycinet and 1 located in Bicheno, approximately 10 kilometres from Freycinet.
(XLSX)

**S8 Table. M5 loss.** (A) Summary of 27 M5 loss events and two M5 gain events in DFT1 biopsies. The M5 loss ID, together with tumour ID, clade group, and information about whether the sample is a cell line, a duplicate, or tetraploid (together with the tetraploid ID, S1 Table), are shown. M5 losses observed in cell lines within this set are inferred to have occurred prior to cell line establishment. M5 loss is defined as having CNVs 6BM (chromosome 2) and 11BM (chromosome 5) and at least one of the chromosome X events, 16R, 17R, 18R. CNV IDs for each event are provided (see S3 Table). Samples in which M5 loss is subclonal are tagged. In tetraploid tumours with polymorphic loss of both M5 copies within the tetraploid group, two M5 loss events are inferred. (B) Summary of 13 M5 loss events occurring exclusively in cell lines in culture or cell lines after inoculation into animals and interpreted as having occurred after cell line establishment. Tumour ID, clade group, and subclonality are indicated, as well as CNV IDs. (C) Observed and expected M5 losses and M5 losses per diploid genome in DFT1 biopsies, cell lines, diploid biopsies, and tetraploid biopsies. Only DFT1s with CNV data are included in the total counts, and duplicates are excluded. "M5 loss ID" refers to parts (A) or (B).
(XLSX)

**S9 Table. Breakpoint reuse.** Coordinates of CNV breakpoints, excluding starts and ends of chromosomes and breakpoints associated with M5. CNV IDs (see S3 Table) associated with the breakpoints and counts of breakpoint use in DFT1, DFT2, and non-DFTD tumours are shown. Three non-DFTD tumours (106T1, 106T2, and 106T3) belong to the same non-DFTD cancer sampled from 3 body sites and breakpoints are collapsed. "Breakpoint" refers to a single coordinate in the genome that was observed to be involved in either a gain or loss CNV. This is different to the data shown in S9 Fig, which shows recurrent cross-cancer CNVs with identical pairs of reused breakpoints.
(XLSX)

**S10 Table. DFT1 cell line CNVs.** Summary of recurrent CNVs associated with DFT1 cell lines. The recurrent CNVs commonly observed in DFT1 cell lines fall into two linked clusters, involving losses on chromosomes 1, 2, and 4 and gains on chromosomes 5 and X, respectively. IDs and coordinates of CNVs part of each cluster are shown, together with genotypes in cell lines and in 18 DFT1 biopsies, which have acquired one or more of the common cell line CNVs. Each independent event, which may be shared between multiple tumours via a common ancestor, has either a "cell line event" or "biopsy event" ID. The tumours associated with each event are listed, together with clade group and metastasis/primary identity if relevant. Tetraploid tumours, together with tetraploid lineage in parentheses, are marked. CNV IDs for each event are shown (see S3 Table). Clonal changes in tetraploid tumours are denoted as subclonal. SC, subclonal; PBM, partial back mutation.
(XLSX)

**S11 Table. Allelic copy number.** (A) Summary of CNVs from DFT1 biopsies and cell lines whose footprint encompassed genotyped MIP alleles (S2 and S3 Tables), as presented in S10 Fig. MIP genotypes and CNVs are only listed in table if the CNV depth in a single direction (gain, loss) was ≥2 and if the MIP ALT allele was either present in the tumour(s) carrying the CNV (gains) or inferred to be present in an ancestor tumour (losses). Details of the sample (clade group, purity), CNV (CNV ID, coordiantes, type (gain, loss), copy number state) and MIP (MIP ID, coordinate, REF and ALT alleles, VAF within the indicated sample) are provided. The best fit genotype (in increments of 0.5) and allele implicated in the CNV are shown. CNVs 3R1 and 9, the parental CNVs of the recurrent chromosome 4 back mutations observed in cell lines, are shown three times each in order to illustrate informative genotyped loci that were captured in different samples. (B) Genotypes at four informative loci in 16 DFT1s (15 cell

lines, 1 biopsy) within the chromosome 4 region repeatedly lost in cell lines (CNV IDs 3R1 and 9). The cell lines have CN2, whereas the biopsy, similar to the DFT1 MRCA, has CN3 at these loci. The 3 chromosomal copies (designated chromosomes A, A1, and B, where A and A1 are somatic duplicates) present in the DFT1 MRCA are shown, and the chromosome inferred to have been lost by each cell line is indicated. The clade group to which each cell line belongs is shown. Samples 87T2 and 90T are a single cell line collected at different passages sequenced independently with identical results. "N," genotype undetermined.
(XLSX)

**S12 Table. Genome coordinates.** Equivalence between genome coordinates used in this study and those in Tasmanian devil reference genome Devil7.1+MT (http://www.ensembl.org/Sarcophilus_harrisii/Info/Index).
(XLSX)

## Acknowledgments

We are grateful to the Wellcome Sanger Institute Core IT team and the Cancer, Ageing and Somatic Mutations IT team for their assistance. We thank volunteers who assisted with Tasmanian devil field work. Robyn Taylor provided technical assistance, and Andrea Strakova, Máire Ní Leathlobhair, Hannah Bender, Anne-Maree Pearse, and Graeme Knowles contributed advice and helpful discussions.

## Author Contributions

**Conceptualization:** Elizabeth P. Murchison.

**Data curation:** Young Mi Kwon, Nicole Potts, Maximilian R. Stammnitz, Rodrigo Hamede, Elizabeth P. Murchison.

**Formal analysis:** Young Mi Kwon, Kevin Gori, Nicole Potts, Naomi Cannell, Elizabeth P. Murchison.

**Funding acquisition:** Rodrigo Hamede, Elizabeth P. Murchison.

**Investigation:** Young Mi Kwon, Kevin Gori, Nicole Potts, Naomi Cannell, Mona Zimmermann, Elizabeth P. Murchison.

**Methodology:** Young Mi Kwon, Kevin Gori, Naomi Park, Jinhong Wang, Maximilian R. Stammnitz, Naomi Cannell, Adrian Baez-Ortega, Mona Zimmermann, David C. Wedge, Elizabeth P. Murchison.

**Project administration:** Michael R. Stratton, Elizabeth P. Murchison.

**Resources:** Kate Swift, Sebastien Comte, Samantha Fox, Colette Harmsen, Stewart Huxtable, Menna Jones, Alexandre Kreiss, Clare Lawrence, Billie Lazenby, Sarah Peck, Ruth Pye, Gregory Woods, David Pemberton, Rodrigo Hamede, Elizabeth P. Murchison.

**Software:** Young Mi Kwon, Kevin Gori, Nicole Potts.

**Supervision:** Menna Jones, Gregory Woods, David C. Wedge, David Pemberton, Rodrigo Hamede, Elizabeth P. Murchison.

**Validation:** Young Mi Kwon, Kevin Gori, Elizabeth P. Murchison.

**Visualization:** Young Mi Kwon, Kevin Gori, Nicole Potts, Naomi Cannell, Elizabeth P. Murchison.

**Writing – original draft:** Elizabeth P. Murchison.

**Writing – review & editing:** Young Mi Kwon, Kevin Gori, Naomi Park, Nicole Potts, Kate Swift, Maximilian R. Stammnitz, Naomi Cannell, Adrian Baez-Ortega, Sebastien Comte, Samantha Fox, Colette Harmsen, Menna Jones, Billie Lazenby, Sarah Peck, Ruth Pye, Gregory Woods, David C. Wedge, David Pemberton, Michael R. Stratton, Rodrigo Hamede, Elizabeth P. Murchison.

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
