## [Editor Report · Decision Letter 0]

13 Jul 2020

Dear Dr Murchison, 

Thank you for submitting your manuscript entitled "Evolution and lineage dynamics of a transmissible cancer in Tasmanian devils" for consideration as a Research Article by PLOS Biology.

Your manuscript has now been evaluated by the PLOS Biology editorial staff, as well as by an academic editor with relevant expertise, and I'm writing to let you know that we would like to send your submission out for external peer review.

Please re-submit your manuscript within two working days, i.e. by Jul 15 2020 11:59PM.

Kind regards,

Roli Roberts

Senior Editor

PLOS Biology

---

## [Decision Letter · Decision Letter 1]

4 Sep 2020

Dear Dr Murchison,

Thank you very much for submitting your manuscript "Evolution and lineage dynamics of a transmissible cancer in Tasmanian devils" for consideration as a Research Article by PLOS Biology. As with all papers reviewed by the journal, yours was evaluated by the PLOS Biology editors as well as by an Academic Editor with relevant expertise and in this case by two independent reviewers.

You'll see that both reviewers are very positive about your paper, but each raises a number of concerns that will need to be attended to. Based on the reviews, we will probably accept this manuscript for publication, assuming that you will modify the manuscript to address the remaining points raised by the reviewers. Please also make sure to address my Data Policy-related requests noted at the end of this email.

We expect to receive your revised manuscript within two weeks. Your revisions should address the specific points made by each reviewer. In addition to the remaining revisions and before we will be able to formally accept your manuscript and consider it "in press", we also need to ensure that your article conforms to our guidelines. A member of our team will be in touch shortly with a set of requests. As we can't proceed until these requirements are met, your swift response will help prevent delays to publication.

- a cover letter that should detail your responses to any editorial requests, if applicable

*Copyediting*

*Published Peer Review History*

*Early Version*

Sincerely,

Roli Roberts

Senior Editor,

rroberts@plos.org,

PLOS Biology

DATA POLICY:

We note that your raw data have been deposited in the ENU. In addition, we ask that all individual quantitative observations that underlie the data summarized in the figures and results of your paper be made available in one of the following forms:

Regardless of the method selected, please ensure that you provide the individual numerical values that underlie the summary data displayed in the following figure panels as they are essential for readers to assess your analysis and to reproduce it: Figs 1ABCDE, 2ABCDEFGHI, 3ABCDE, S1, S2, S4, S5, S6, S7, S8ABCDEFGH, S9ABC, S10, S11, S12AB. We see that you have deposited some data in Zenodo, but its relationship to the Figures is unclear, and we will need to check it before publication. Please could you clarify the relationship, and also send us a password or reviewer link so that we can assess it? NOTE: the numerical data provided should include all replicates AND the way in which the plotted mean and errors were derived (it should not present only the mean/average values).

REVIEWERS' COMMENTS:

Reviewer #1:

Mi Kwon et al. is a very interesting study of the unique transmissible cancer in Tasmanian devils. The paper has two main parts: The first uses genomic data to show the evolutionary history of DFT1 and how the clades have spread across the population and replaced each other in different populations multiple times. The second analyzes the copy number changes over time in order to better understand the genetic stability of transmissible cancer. This study analyzed a massive number of DFT1 samples and provides a significant increase in our knowledge of DFT1 evolution at both the ecological level and the level of genetic and chromosomal stability. It will be of broad interest to researchers interested in biology, cancer, ecology, and pathogens. I have several minor questions and comments below. 

1. Page 4, line 4: "Remarkably, of the one hundred devils hosting two or more DFT1 tumours within the cohort, 37 had tumours with 'public' genotypes shared between tumours in more than one individual (Figure 1E and Table S4). Most of these likely represent co-infection, but in some cases we may have sampled the source of a new genotype." Is this exactly 100 or about 100? The exact number should be here, and I just want to double check as 100 is a very round number. Also, the number of total cases considered here should be shown in the text and/or figure legend (100 out of how many?) When making a conclusion about the rate of coinfection it could very easily be misinterpreted as saying that 37% of cases are co-infection. The actual rate of coinfection is important and relevant in many real ways and it should be stated, while the % of devils carrying two tumours that are co-infected is not. The description of "public" genotypes is confusing here, and could be clarified. This is explained well in the figure legend. Perhaps the two sentences in the figure legend could be moved to the main text.

2. Page 4, line 26 and Figure 1, I would love to know where the tetraploid cancers fit within this evolutionary tree and geographic spread! The data in Figure S4 show the presence of the tetraploids over time, but do not show their prevalence. Is there any evidence that tetraploid subclades were successfully able to replace diploid strains or were they short-lived in all cases? Due to the possible role of polyploidy in human cancer it would be of interest to know whether tetraploidy in DFT1 was ever able to out-compete diploid cancers or whether it was less successful in all cases.

3. Figure 2A. How are tetraploids dealt with in this analysis? A line about this could be added to the legend.

4. Page 4 line 28 and Figure 2B, What are the A-E and A-D ancestor genotypes? In the text you describe the creation of a single ancestor. It appears you have made 2 ancestors, one an ancestor of clade A and E, and one from clade A and D. The A-E clade is clearly earlier and contains all DFT1 samples, and has an interesting lack of the loss at B2M. The A-D ancestor is unclear. Due to the ambiguous placement of D in the tree, it is unclear whether this ancestor includes clade B or C or both. If it includes all DFT1 samples except E that would be useful to state. 

5. Page 4, line 37. The text refers to Figure 2B lower panel as an example of a cancer with a high CNV burden, but that panel appears to show a tetraploid cancer with minor additional changes compared to the diploid example. Was this a mistake? Was there an additional example that was supposed to be here?

6. Figure 2C. Are these CNVs relative to the normal devil reference or relative to a DFT ancestor?

7. Page 5, line 15. The number of occurrences of a gain or loss is not necessarily evidence of the selection of cancers with these events. Were these independent events found in a single animal or was there evidence of their spread? This could be critical evidence of selection for these events. For example, M5 has been lost multiple times, but the authors do not consider it to be positively selected—this is hypothesized to be an unstable locus with no selection for or against. The finding of unstable recurrent sites of mutation in Figure 2I also argue that this may not be due to selection. I think selection is an interesting question and might be addressable with the data here, but it has not been shown. Also, how was the determination made that a gain or loss was independent? Was this based on determination of the breakpoint junction or based on phylogenetic analysis?

8. Figure 2F. What are the statistical tests comparing? In particular, is it testing whether the M5 losses in clade B are lower than expected, or that clade A2 are higher, or that overall the patterns of predicted and observed are different? And expected based on what? Presumably expected based on the total number of losses observed and the frequency of observation of the ploidy/clade? This could be succinctly mentioned in text or legend. 

9. Figure 2H. Are the number of CNVs for DFT1 and DFT2 samples comparing each DFT sample with the reference or identifying unique CNVs for each sample or relative to the DFT ancestors? It seems like they are total CNVs relative to the reference, but that is suggesting that a primary carcinoma in a single devil has more CNVs than DFT1 has in its total >20 year history.

10. Page 6, line 22. This analysis is unclear. "One hundred and nineteen CNV breakpoints (at 100 kb resolution) were reused across two or more independent devil cancers" Does this refer to the non-DFT cancers? Or is this including non-DFT cancers and DFT1 and 2? Or "independent" DFT samples? I think the first, but it is not clear. In Fig 2I it appears that the first two bars a samples DFT1 and DFT2 respectively, but maybe they instead represent simulated and observed data from a DFT1/DFT2 comparison. But the column for DFT1 in the DFT1/non-DFTD dataset appears to have blue in it, so it is not just simulated data, so I cannot quite figure it out. Additionally, since there are so many independent cases of CNV at the same sites in independent cancers, have the authors considered that the CNV might be a germline polymorphism in the population, rather than a recurrent somatic mutation. 

11. Figure S9. In the legend the authors state: "No CNVs occurring independently in more than one non-DFTD tumour, or in one or more DFT1, DFT2 and non-DFTD tumour were found." This statement occurs in the legend of the supplementary figure that is describing the recurrent CNVs with identical breakpoints. Figure 2G and 2I both appear to show CNVs occurring independently. That was a major conclusion of that section in the paper. Is there a typo in this sentence or am I totally misunderstanding the authors' statements? This needs to be clarified.

12. Page 6, line 30, "Laboratory cell lines can be readily established from DFT1 tumours." Is there a reference for this?

13. Page 6, line 36, "Altogether, this suggests that introduction to cell culture may select for genetically unstable DFT1 subclones." Rather than selection for a greater rate of mutation, it seems like an equally likely hypothesis that there has been a stable rate of mutation and relaxation of negative selection for maintenance of genes required for growth and spread in the host. The genes not needed for survival in a dish, but needed for survival in a host would be a very interesting question to pursue in the future. 

14. Also, what was the outcome of the 9 cultured DFT lines that were inoculated into live devils? How long did the cells grow before they were screened for the presence of the CNVs? These details should be included somewhere.

Reviewer #2:

Young et al present a very interesting study on the evolution and spread of a transmissible tumour in Tasmanian devils. This unusual tumour provides a fascinating system in which to study tumour evolution and the authors have done an excellent job of uncovering how the tumour has changes as the disease spread across Tasmania. The paper is well written and the extensive analyses performed conveniently displayed in an easy to understand yet comprehensive figures. I commend the authors on making their findings so accessible for a broad audience. I also commend the authors on the inclusion of an examination of tumour evolution under cell culture conditions vs tumour biopsies. Overall, this is an impressive study. I only have one minor comment that the authors many wish to address: 

When I looked at Table S1, I noticed that there was information for a portion of the samples on the tumour strain. I think it would be useful to point out that the clades identified in this study do not match the karyotypic strains identified in Pearse et al 2012 and why this might be. The presence/absence of M5 was used as a point of difference between strains 1 and 2 but the results present from the sequence analyses indicates that M5 has been lost multiple times and isn't indicative of a different tumour "strain".

---

## [Editor Report · Decision Letter 2]

19 Oct 2020

Dear Dr Murchison,

On behalf of my colleagues and the Academic Editor, Matt van de Rijn, I am pleased to inform you that we will be delighted to publish your Research Article in PLOS Biology. 

Early Version

PRESS 

Kind regards,

Alice Musson, 

PLOS Biology

on behalf of

Roland Roberts,

Senior Editor

PLOS Biology